# A secondary mechanism of action for triazole antifungals in *Aspergillus fumigatus* mediated by *hmg1*

Jeffrey M. Rybak [1], Jinhong Xie [2,3], Adela Martin-Vicente[3], Xabier Guruceaga [3], Harrison I. Thorn [2,3], Ashley V. Nywening[3,4,5], Wenbo Ge[3], Ana C. O. Souza [1], Amol C. Shetty [6], Carrie McCracken[6], Vincent M. Bruno[6,7], Josie E. Parker [8], Steven L. Kelly [9], Hannah M. Snell [10], Christina A. Cuomo [10], P. David Rogers[1,11] & Jarrod R. Fortwendel [3,5,11] ✉

Triazole antifungals function as ergosterol biosynthesis inhibitors and are frontline therapy for invasive fungal infections, such as invasive aspergillosis. The primary mechanism of action of triazoles is through the specific inhibition of a cytochrome P450 14-α-sterol demethylase enzyme, Cyp51A/B, resulting in depletion of cellular ergosterol. Here, we uncover a clinically relevant secondary mechanism of action for triazoles within the ergosterol biosynthesis pathway. We provide evidence that triazole-mediated inhibition of Cyp51A/B activity generates sterol intermediate perturbations that are likely decoded by the sterol sensing functions of HMG-CoA reductase and Insulin-Induced Gene orthologs as increased pathway activity. This, in turn, results in negative feedback regulation of HMG-CoA reductase, the rate-limiting step of sterol biosynthesis. We also provide evidence that HMG-CoA reductase sterol sensing domain mutations previously identified as generating resistance in clinical isolates of *Aspergillus fumigatus* partially disrupt this triazole-induced feedback. Therefore, our data point to a secondary mechanism of action for the triazoles: induction of HMG-CoA reductase negative feedback for downregulation of ergosterol biosynthesis pathway activity. Abrogation of this feedback through acquired mutations in the HMG-CoA reductase sterol sensing domain diminishes triazole antifungal activity against fungal pathogens and underpins HMG-CoA reductase-mediated resistance.

*Aspergillus fumigatus* is the major etiologic agent of invasive aspergillosis (IA), a severe and life-threatening invasive fungal infection impacting more than 300,000 people per year[1]. Even with timely diagnosis and administration of appropriate antifungal therapy, the mortality rates for IA often exceed 50%[1]. The frontline antifungals used to combat IA are members of the triazoles, an antifungal class of compounds that inhibit 14α-sterol-demethylase activity to deplete cellular sterols and, for many fungal species, causes the accumulation of toxic sterol intermediates that disrupt the cell membrane[2]. The triazoles are the most widely used antifungal class and, for treatment of IA, include four drugs employed clinically that each exhibit fungicidal anti-*Aspergillus* activity: voriconazole, itraconazole, posaconazole, and isavuconazole[3]. Although they are considered highly effective, there is a growing appreciation that triazole-based therapy

for IA is negatively impacted by a recent rise in resistance noted globally among clinical isolates of *A. fumigatus*[4–7].

How resistance to triazoles develops in environmental organisms like *A. fumigatus* has been the focus of intense study for the past two decades. Although the major mechanism of resistance involves mutations in the gene encoding the *A. fumigatus* cytochrome P450 triazole target, *cyp51A*, our group, and others, have also identified mutations affecting the HMG-CoA reductase (HMGCR) enzyme, Hmg1, as underpinning *A. fumigatus* clinical resistance to triazole antifungals[8,9]. Since this initial discovery, mutations in Hmg1 have been noted in resistant *A. fumigatus* clinical isolates from around the world[10–16]. In eukaryotic organisms, HMGCR functions as a rate-limiting step in the mevalonate biosynthesis pathway, a sequence of reactions producing farnesyl pyrophosphate as an intermediate for cholesterol biosynthesis in mammalian cells or for ergosterol biosynthesis in fungi. HMGCR orthologs are endoplasmic reticulum (ER)-resident, multi-domain proteins made up of an N-terminal portion that encodes a conserved sterol sensing domain (SSD), a linker domain, and a C-terminal catalytic domain[17]. Although the exact mechanism of how HMGCR mutations in *A. fumigatus* isolates cause resistance is currently unclear, the resistance-driving mutations identified thus far are all contained within the predicted SSD.

The HMGCR SSD is composed of multiple transmembrane spanning regions and plays important roles in feedback regulation of the HMGCR enzyme. In mammalian cells, the SSD is required for sensing accumulation of specific isoprenoids and sterols that are intermediates of the cholesterol biosynthesis pathway[17–19]. Accumulation of these cholesterol pathway intermediates, specifically geranylgeranyl pyrophosphate and lanosterol, induces negative feedback control of pathway activity by promoting binding of HMGCR to another ER-resident protein, INSIG (Insulin Induced Gene)[19]. INSIG binding to HMGCR recruits enzymes that ubiquitinate HMGCR promoting its turnover through ER-Associated Degradation (ERAD)[18,20]. Through this mechanism, the SSD functions to detect sterol intermediates as a readout for pathway activity and, in turn, downregulates HMGCR protein abundance when pathway activity appears higher than required (i.e., overaccumulation of sterol and isoprenoid intermediates). INSIG-mediated HMGCR regulation differs somewhat in yeast organisms. In *Saccharomyces cerevisiae*, accumulation of sterol intermediates also imparts negative-feedback regulation of HMGCR that is primarily enacted through induction of accelerated degradation[17,21]. However, the INSIG ortholog in budding yeast, Ins1p, appears to inhibit this feedback mechanism rather than promote it[17]. Similar to the mammalian system, the INSIG ortholog of the fission yeast, *Schizosaccharomyces pombe*, Ins1, promotes negative feedback of HMGCR[17]. The mechanism of Ins1-mediated feedback regulation in *S. pombe* is not achieved through accelerated degradation but rather through promotion of an inactivating phosphorylation event in the HMGCR catalytic domain[17,22,23]. Regardless of the exact mechanism employed (i.e., accelerated degradation vs. phospho-regulation), a conserved function of the SSD is to sense accumulation of pathway intermediates to induce negative feedback on HMGCR. This feedback is typically modulated by the INSIG orthologs that work to either promote or inhibit the process.

We previously hypothesized that SSD mutations in triazole-resistant isolates of *A. fumigatus* may disrupt the conserved sterol sensing functions of the SSD. The resulting loss-of-function may partially block sensing of negative feedback signals that are likely produced by accumulation of sterol intermediates caused by triazole-induced inhibition of the 14α-sterol-demethylase, Cyp51A/B. Here, we show through analysis of an expanded clinical isolate library and direct genetic testing that triazole resistance is only found to be associated with mutations impacting the predicted Hmg1 SSD. We further show that SSD mutation-mediated resistance is unique to the activity of triazole antifungals and not to other mevalonate or ergosterol biosynthesis pathway inhibitors. Finally, we show that this resistance is dependent on the INSIG ortholog, InsA, but is independent of the conserved ERAD E3 ubiquitin ligase, HrdA. Our findings support a model wherein triazole-mediated inhibition of Cyp51A/B generates accumulation of important sterol intermediates, likely lanosterol, that are decoded by the Hmg1 SSD as increased pathway activity. This signal, in turn, induces negative feedback regulation of Hmg1 through InsA that is likely not mediated through accelerated degradation. Thus, exploration of *hmg1* SSD mutation-mediated resistance has revealed a previously unknown and an underappreciated secondary mechanism of action of the triazole antifungals: induction of negative feedback on HMG-CoA reductase.

## Results

### Hmg1-mediated triazole resistance is unique to variations within the sterol sensing domain

To test the hypothesis that mutations specifically within the Hmg1 SSD were causative of triazole resistance, we first sought to expand on our previous analysis[8] by employing a genomic dataset from 87 *A. fumigatus* clinical isolates carrying *hmg1* genetic variants to identify and characterize additional Hmg1 mutations driving clinical resistance (Supplementary Data 1). In addition to the previously identified mutations[8], our analyses revealed frequently mutated sites outside the SSD residing in the fungus-specific N-terminal region, the linker domain, and the catalytic region of the Hmg1 protein (Fig. 1). In addition, there were multiple sites that appeared to be mutated within the SSD of multiple isolates, with recurrent amino acid changes observed across isolates (Fig. 1). Although variants could be identified in both triazole-resistant and -susceptible isolates, it is notable that all 26 isolates found to have *hmg1* mutations altering residues within the predicted SSD (30% of the 87 isolates analyzed) were resistant to at least one anti-*Aspergillus* triazole (Fig. 1, green box), and 24 of these 26 isolates (92%) were characterized as being resistant to multiple triazoles. Additionally, 10 of the 26 SSD mutant isolates (38%) were found to lack non-synonymous mutations in the *cyp51A* gene that could alternatively or additionally explain resistance (Fig. 1). Recent work has described the triazole resistance-associated *cyp51A* TR34/L98H mutation as largely specific to one of two-to-three major *A. fumigatus* clades[24,25]. To see if Hmg1 SSD mutations in our isolates appeared only in specific clades, we next matched our isolate phylogeny to that reported in a recent study by Rhodes et al. that defined two major *A. fumigatus* clades[24]. This analysis employs the frequency of the *cyp51A* TR34/LH98 mutation within the population to assign isolates to either Clade A or Clade B. As noted in Fig. 1, we uncovered a clear division wherein the first major population of our isolates, DI15-105 through F18085, corresponded to Clade B and isolates C60 through C36 corresponded to Clade A. Although the majority of isolates carrying Hmg1 SSD mutations were assigned to Clade B, this relationship was not exclusive as multiple Clade A isolates were also found to carry Hmg1 SSD mutations (Fig. 1). Using this approach, four isolates, DI-20-87, DI-20-130, DI-20-128, and DI-120-100, were classified with less clarity, mapping between Clades A and B (Fig. 1). As previously noted, however, another recent study reported three major *A. fumigatus* clades, rather than two, which could explain this result[25].

To directly test mutations affecting each of the major Hmg1 protein domains, a total of ten non-synonymous variants, most associated with resistant isolates, were selected for genetic validation of their potential to influence Hmg1-mediated resistance (Fig. 2A). The E105K, D242H, W272L, C402R, V403D, G483R, L558R, and V995I variants were each identified as the sole Hmg1 mutation in, at least, one of the isolates analyzed in this study. Whereas these variants were each validated for their individual contributions to resistance, combination variant *hmg1* alleles were also identified and were validated. For

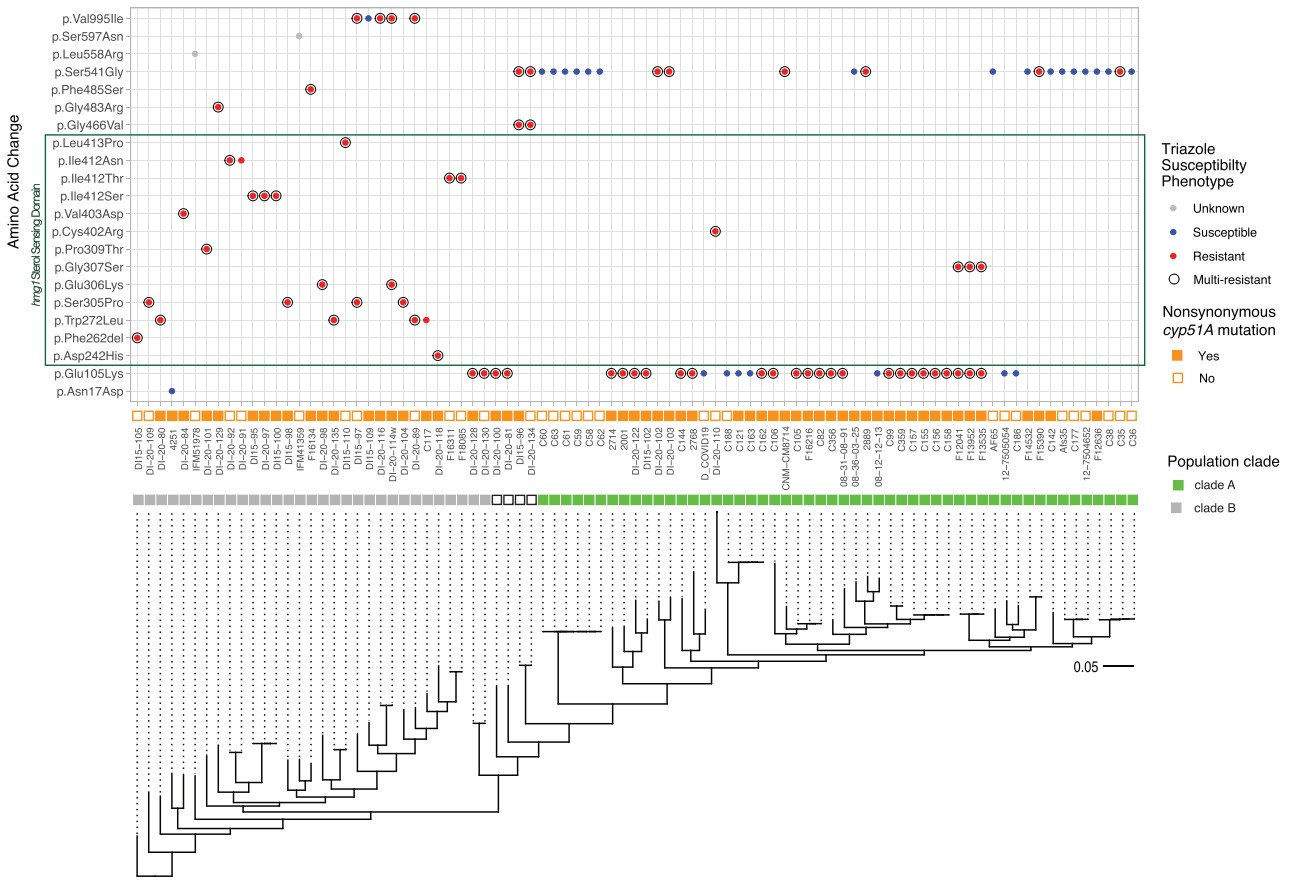

**Fig. 1 | Variant analysis of the *hmg1* allele across clinical isolates identifies mutations associated with triazole resistance.** Mutations residing within the predicted sterol sensing domain (SSD) are highlighted in the green box. Triazole susceptibility phenotype, *cyp51A* genotype, phylogenetic relationships, and clade grouping of the clinical isolate set are indicated.

example, the V995I variant located within the Hmg1 catalytic domain was identified alone and in combination with a previously unvalidated transmembrane domain II (TMDii) SSD variant, W272L, as well as with a TMDiii SSD variant we previously characterized as driving resistance, S305P (Fig. 2A)[8]. In addition, the linker region variant S541G was identified both as a sole mutation, as well as a combination mutation with G466V (Fig. 2A). Variants were each introduced into the triazole susceptible laboratory strain, CEA10, by a previously described "allele swap" technique that utilized CRISPR/Cas9 gene editing[8,26]. Each resulting strain expresses the introduced gene as the sole copy of *hmg1* with expression driven by the endogenous promoter. A manipulation control, wherein the wild-type *hmg1* allele was re-introduced in the same manner [*hmg1*^WT], resulted in no changes to growth, morphology, or triazole susceptibility when compared to the CEA10 parental strain (Fig. 2B–E). Strikingly, we found only those variants that impacted the transmembrane spanning portions of the SSD and linker region were able to generate resistance to mold-active triazoles when expressed in CEA10. Depending on the triazole utilized, we noted 2- to 8-fold increased MIC values for the D242H, W272L, C402R, V403D, and G483R variants (Fig. 2B–E). In addition, the V995I/W272L and V995I/S305P combination variants, but not the V995I mono-variant, were able to generate pan-azole resistance (Fig. 2B–E). These data suggest that amino acid variations specifically in the predicted SSD, as well as those within the adjacent linker domain TMD (which may also participate in sterol sensing in *A. fumigatus*) are uniquely causative of Hmg1-mediated triazole resistance. Interestingly, six of the eight clinical isolates from our collection that carry these *hmg1* mutations display

triazole MIC values that are within a single two-fold dilution of the mutant strains we constructed (Fig. 2 and Supplementary Data 1). Therefore, the *hmg1* mutations within those isolates likely account for the majority of the resistance detected.

## Hmg1 SSD mutation alters triazole-induced *erg* gene expression and sterol accumulation

We previously reported that SSD mutation results in altered sterol intermediate profiles compared to wild-type strains in the absence of triazole stress[8]. However, Cyp51A/B inhibition induces accumulation of sterol intermediates, including lanosterol, and some of these ergosterol precursors may serve as important feedback signals for SSD detection and subsequent pathway regulation[8,19,27,28]. Therefore, we employed the Hmg1^F262del, Hmg1^S305P, and Hmg1^I412S mutant strains generated in our previous study[8] to perform mevalonate and ergosterol biosynthesis pathway gene expression, as well as sterol profiling analyses, to determine if the response to voriconazole was dependent on the function of an intact Hmg1 SSD. Strains were first cultured for 48 hrs in the presence or absence of 0.5x MIC voriconazole and RNA was extracted for RNAseq analysis. Differential gene expression analysis was performed and genes that were significantly upregulated (2-fold or more) in common among the SSD mutants versus controls were analyzed by GO analysis using FungiFun (https://elbe.hki-jena.de/fungifun/). Among the genes that were found upregulated in common between the mutants versus the control strain in the absence of voriconazole treatment (*n* = 175), significantly enriched GO categories included: fatty acid biosynthetic process (*n* = 6), response to stress

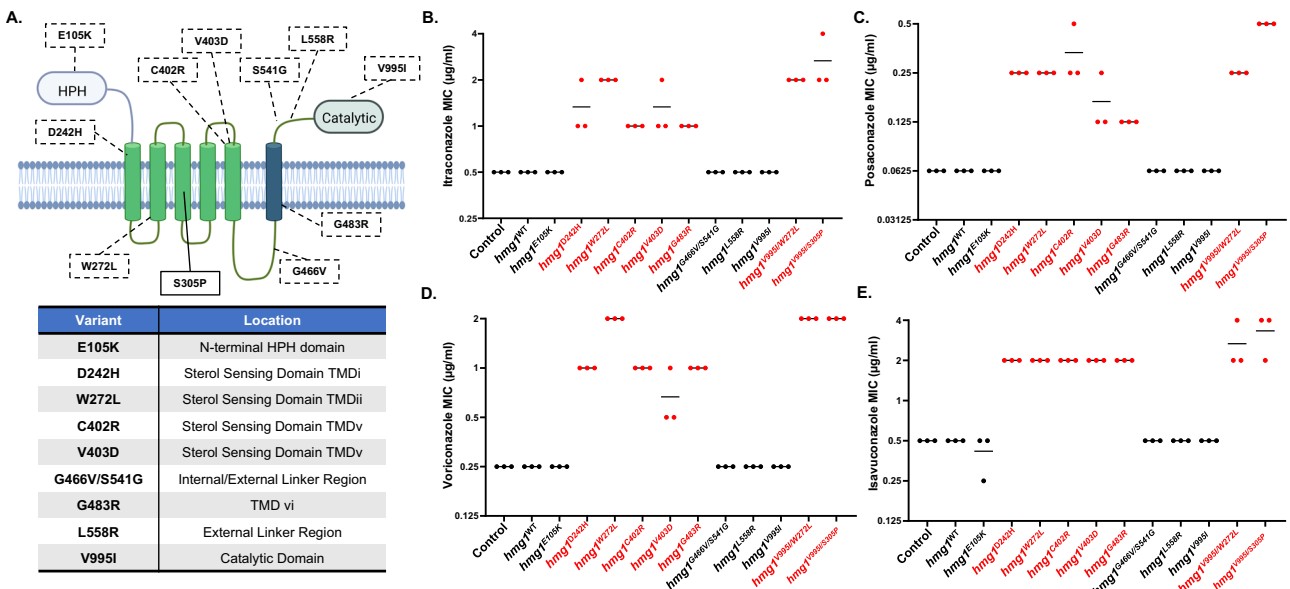

**Fig. 2 | Hmg1-mediated triazole resistance is uniquely driven by mutations in the predicted sterol sensing domain. A** Schematic of the protein domain structure of *A. fumigatus* Hmg1. The location of mutations identified in genome sequences selected for this study are indicated. The previously characterized S305P mutation is indicated for context. Illustration prepared using BioRender.com. Susceptibility profiles of the parental control strain (Δ*akuB-pyrG⁺*), the manipulation control strain (*hmg1*^WT), and the indicated *hmg1* mutants to mold-active triazoles are shown for intraconazole (**B**), posaconazole (**C**), voriconazole (**D**), and isavuconazole (**E**). CLSI-based minimum inhibitory concentration (MIC) assays were performed in triplicate for each strain. Red font indicates mutant strains with shifts in susceptibility compared to controls.

($n = 5$), oxidation-reduction process ($n = 31$), nitrate assimilation ($n = 2$), and oxidoreductase activity ($n = 28$) (Supplementary Data 2). Within the fatty acid biosynthesis GO category and among an exhaustive list of mevalonate and ergosterol pathway genes, *erg24A*, *erg24B*, *erg25A*, and *erg3B* were the only ergosterol biosynthesis members identified as upregulated 2-fold or more (Fig. 3A, B and Supplementary Data 2). Of the 104 significantly upregulated genes (2-fold or more) that were identified in common among the SSD mutants versus the control strain in response to triazole stress, significantly enriched GO categories included: secondary metabolite biosynthetic process ($n = 5$), cellular component ($n = 75$), oxidoreductase activity ($n = 20$), response to stress ($n = 4$), and oxidoreduction process ($n = 21$). Although not identified in the enriched GO categories, the C24 sterol-methyltransferase, *erg6*, was identified as significantly upregulated in common in the SSD mutants specifically in response to triazole stress (Fig. 3B and Supplementary Data 2).

In the absence of voriconazole treatment, 54 genes were found commonly downregulated (0.5-fold or less) in the Hmg1 SSD mutants when compared to the control strain (Supplementary Data 2). Of those, significantly enriched GO categories included: oxidoreductase activity (14), oxidation-reduction process (14), transferase activity (10), catalytic activity (9), mycotoxin biosynthetic process (7), heme binding (6), gliotoxin biosynthetic process (5), fumagillin biosynthetic process (5), secondary metabolite biosynthetic process (5), secondary metabolic process (5), monooxygenase activity (5), iron ion binding (5), oxidoreductase activity, acting on paired donors, with incorporation or reduction of molecular oxygen (4), pathogenesis (3), nonribosomal peptide biosynthetic process (2), ribosomal small subunit assembly (1), positive regulation of pseurotin A biosynthetic process (1), positive regulation of fumagillin biosynthetic process (1), sulfate assimilation, phosphoadenylyl sulfate reduction by phosphoadenylyl-sulfate reductase (thioredoxin) (1), phosphoadenylyl-sulfate reductase (thioredoxin) activity (1), lanosterol synthase activity (1), (17Z)-protosta-17(20),24-dien-3beta-ol biosynthetic process (1) and cytosolic small ribosomal subunit (1). Under triazole drug stress, 36 genes were found commonly

downregulated in the SSD mutants, and no GO categories were significantly enriched.

To determine whether these transcriptional alterations translated into sterol accumulation, strains were grown as above, and sterols were extracted for GC/MS analysis. Using a similar approach, we previously reported a significant decrease in the relative proportion of ergosterol, along with increases in the proportions of several ergosterol precursors, in the Hmg1 SSD mutant strains under non-stress conditions[8]. We also previously noted that the relative decrease in the proportion of ergosterol in mutant strains was not correlated with a decrease in the total ergosterol content (μg ergosterol/mg dry weight)[8]. Here, we acquired similar results in untreated Hmg1 SSD mutant strains, with relatively minor accumulation of 4,4 dimethyl Ergosta 8,24(28), Fecosterol, and Ergosta 5,7,24(28) sterol intermediates (Fig. 3A–C and Supplementary Data 3). These altered profiles suggested basal activity of Erg24A/Erg24B, Erg25A/Erg25B, and Erg3B, respectively, may be slightly elevated in the Hmg1 SSD mutants, a finding that correlated well with increased transcription of these genes that were noted in the RNAseq profiles (Fig. 3A–C and Supplementary Data 3). As expected, voriconazole treatment caused a significant increase in the relative proportion of both lanosterol and eburicol, the sterol intermediates directly upstream of Cyp51A/B and Erg6 (Fig. 3A, C and Supplementary Data 3). In addition, the relative increases in lanosterol and eburicol induced by voriconazole were similar between the control and Hmg1 SSD mutant strains. In contrast, whereas both the control and mutant strains displayed a relative increase in C-28 methylated sterols under voriconazole stress, the Hmg1 SSD mutant strains uniquely exhibited an exaggerated accumulation of these sterol derivatives (Fig. 3C and Supplementary Data 3). This result suggested that frequency of methylation events mediated by the sterol methyltransferase, Erg6, may increase in response to voriconazole-mediated inhibition of the Cyp51A/B and that this effect is further amplified by SSD mutation. These findings again correlated well with our transcriptional data denoting increased expression of *erg6* in response to voriconazole, especially in the Hmg1 SSD mutant strains (Fig. 3B). Finally, although voriconazole treatment caused a decrease in the

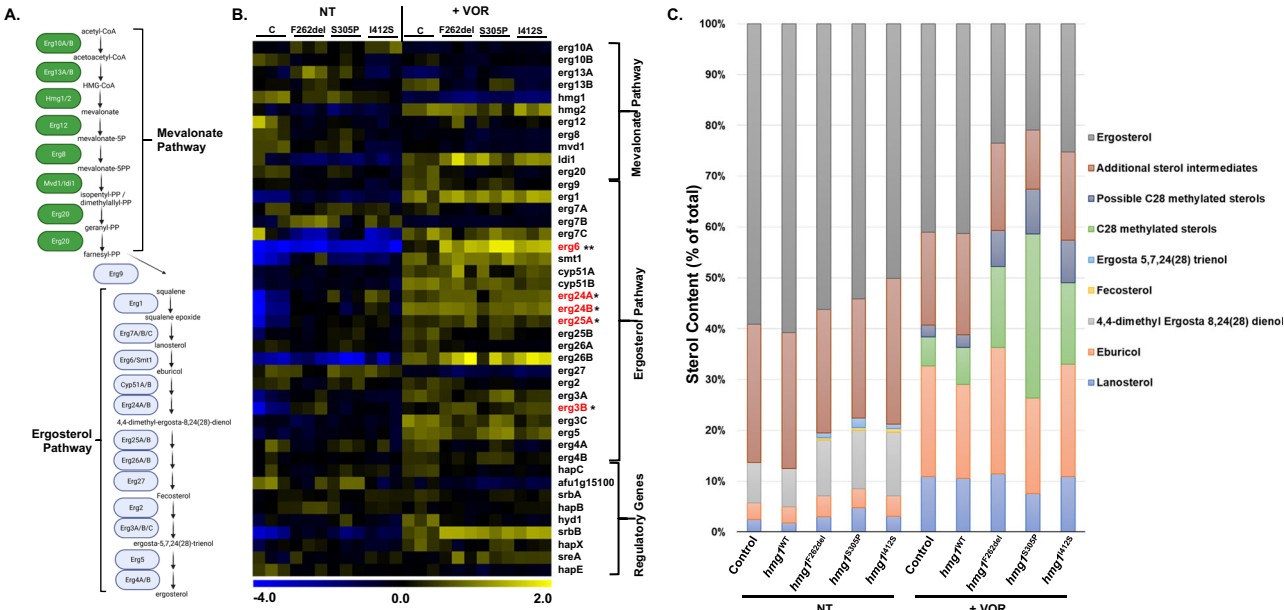

**Fig. 3 | Hmg1 SSD mutation results in altered sterol biosynthesis in response to voriconazole. A** Schematic of the mevalonate and ergosterol biosynthesis pathways in *A. fumigatus*. Intermediate sterols of the ergosterol pathway are indicated for their relevance to RNAseq and sterol profiling analysis results. Illustration prepared using BioRender.com. **B** Heat map representing differential gene expression analysis of known and predicted mevalonate and ergosterol biosynthesis pathway genes of the parental control (**C**), *hmg1*^F262del (F262del), *hmg1*^S305P (S305P), and *hmg1*^I412S (I412S) strains. All strains were cultured in the vehicle (NT) or 0.5 X MIC voriconazole (+VOR) for 48 hours at 37 °C with shaking at 250 rpm and differentially expressed genes were determined by comparing each SSD mutant to the respective control under NT or VOR conditions. * = significantly upregulated 2-fold or more in common among the Hmg1 SSD mutant strains under NT conditions. ** = significantly upregulated 2-fold or more in common among the Hmg1 SSD mutant strains under +VOR conditions. **C** Relative sterol distribution in the control and Hmg1 SSD mutant strains. Strains were cultured as in **B** and extracted sterols analyzed by GC-MS.

relative proportion of ergosterol in all strains, we noted a greater decrease in the Hmg1 SSD mutants (Fig. 3C and Supplementary Data 3). However, as we previously uncovered, we again found that these proportional differences were not driven by changes in total ergosterol content (µg ergosterol/mg dry weight) under non-stress conditions. Therefore, we interpret this to be due to the increased proportion of multiple ergosterol precursor derivatives in the Hmg1 SSD mutants. Together these results indicate that an intact Hmg1 SSD is required to limit the accumulation of sterol intermediate derivatives during triazole-mediated inhibition of Cyp51A/B.

## Hmg1 SSD mutation or mutations that alter the transcriptional balance of *hmg1*-to-INSIG uniquely alter susceptibility to triazoles

We next sought to solidify a role for the SSD in triazole stress responses and to pinpoint steps in the ergosterol pathway that potentially drive Hmg1-regulating sterol accumulation. To do this, we utilized SSD mutant strains generated from this study [*hmg1*^W272L and *hmg1*^C402R] in combination with mutant strains from our previous work [*hmg1*^F262del, *hmg1*^S305P, and *hmg1*^I412S][8] that now have genetically validated roles in triazole resistance to perform susceptibility assays against a panel of mevalonate and ergosterol pathway inhibitors. These inhibitors included: the HMGCR statin class inhibitor, rosuvastatin; the squalene epoxidase inhibitor in the antifungal allylamine class, terbinafine; and the front-line anti-*Aspergillus* triazole, voriconazole (Fig. 4A). Equal numbers of conidia from each strain were inoculated onto RPMI agar medium embedded with ascending concentrations of each mevalonate and ergosterol biosynthesis pathway inhibitor and plates were incubated for 72 hours at 37 °C. When compared to wild type and manipulation control strains, no change in susceptibility was evident in the presence of rosuvastatin for any Hmg1 SSD mutant strain as complete growth inhibition was achieved at 80 µg/ml (Fig. 4B). This finding is fitting with the mechanism of HMGCR

inhibition by statins, which involves binding to the C-terminal catalytic domain to block enzyme activity rather than the SSD[29]. For the ergosterol biosynthesis pathway inhibitors, whereas all strains displayed similar susceptibilities to terbinafine (Fig. 4C), the SSD mutants consistently displayed a 4- to 8-fold increase in resistance to voriconazole when compared to control (Fig. 4D). Together, these findings suggest that the Hmg1 SSD may be particularly tuned to sensing sterols accumulating between the squalene epoxidase and 14-α-sterol-demethylase steps.

As our data implies Hmg1 SSD mutation-mediated resistance was unique to triazole antifungals among the pathway inhibitors we employed, we next sought to confirm that triazole-induced negative feedback of Hmg1 through INSIG was the likely mechanism. In mammalian systems, overexpression of either HMGCR or INSIG has been shown to disrupt feedback[20,30]. To examine if a similar INSIG-to-HMGCR regulatory relationship is conserved in *A. fumigatus*, we performed promoter replacement experiments to overexpress either *hmg1* or the sole *A. fumigatus* INSIG ortholog, *insA*, in the wild-type background. The endogenous promoter for either gene was replaced with the strong *hspA* promoter using CRISPR/Cas9 gene editing (Fig. 5A), as we previously described[8]. Gene expression analyses revealed that this manipulation resulted in a more than 50-fold upregulation of each gene (Fig. 5B, C). Two independent overexpression mutants for each gene were examined for their ability to survive rosuvastatin, terbinafine, or voriconazole stress. Overexpression of *hmg1* [*hmg1*^pHspA] induced resistance to the on-target inhibitor, rosuvastatin, indicating that our genetic manipulation results in increased abundance of the Hmg1 target protein (Fig. 5D). This finding is also again in agreement with the mechanism of action of statin drugs, which directly bind to the catalytic region and competitively inhibit HMGCR enzymes in an SSD-independent fashion[29]. When comparing effects of the ergosterol biosynthesis pathway inhibitors, the *hmg1*^pHspA mutants displayed resistance to the triazole antifungals but not to

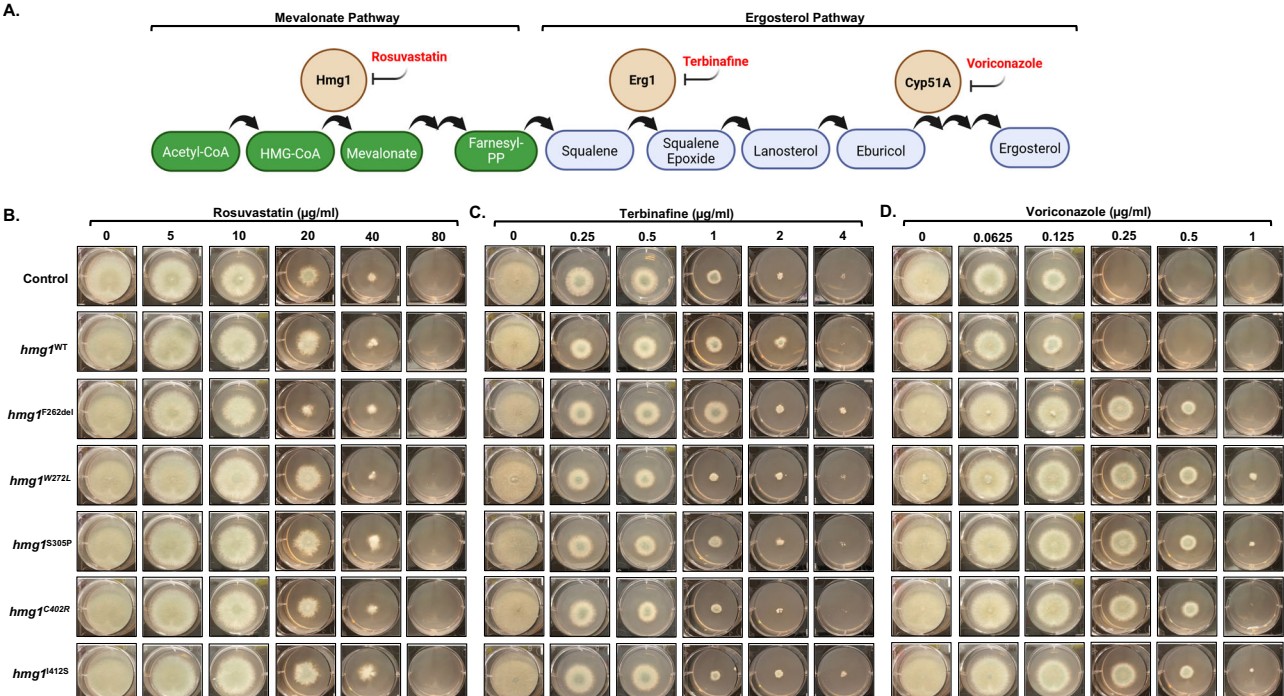

**Fig. 4 | Mutation of the Hmg1 SSD specifically alters response to triazoles and not allylamines or statins. A** Schematic of the mevalonate and ergosterol biosynthesis pathways highlighting enzymatic steps inhibited by statins (rosuvastatin), allylamines (terbinafine), and triazoles (voriconazole). Illustration prepared using BioRender.com. Radial growth assays of the parental and manipulation controls and the Hmg1 SSD mutants to assess susceptibility changes are shown for rosuvastatin (**B**), terbinafine (**C**), and voriconazole (**D**). Conidia (5 × 103) from each strain were spot inoculated onto RPMI media (0.2% glucose, pH 7.0) containing the indicated amount of drug and plates were cultured for 72 hours at 37 °C. Source data are provided as a Source Data file.

terbinafine (Fig. 5E and F), coinciding with data from a separate recent report[31]. This was true for all mold-active triazoles (Supplementary Fig. 1A). The level of triazole resistance in the *hmg1*^pHspA strains (~4- to 8-fold increased MIC) was similar to that noted for the Hmg1 SSD mutants (compare to Fig. 4E). Strikingly, overexpression of *insA* [*insA*^pHspA] generated the opposite phenotype in response to triazoles. The *insA*^pHspA mutants exhibited 2- to 8-fold decreased MIC to each mold-active triazole (Fig. 5F and Supplementary Fig. 1B). In contrast, *insA* overexpression did not alter terbinafine or rosuvastatin susceptibility (Fig. 5D and E). Therefore, the InsA-to-Hmg1 regulatory relationship is conserved in *A. fumigatus* and is important for normal triazole stress response.

## Overexpression of InsA does not resolve Hmg1 SSD-mediated triazole resistance

To see SSD mutation disrupts InsA-mediated regulation, we next overexpressed *insA* in the *hmg1*^pHspA and *hmg1*^S305P genetic backgrounds. We reasoned that, if mutations within the SSD were inhibiting sterol sensing and, in turn, InsA-mediated negative feedback, then our SSD mutants (i.e., *hmg1*^S305P) should be insensitive to the effects of decreasing the Hmg1-to-InsA balance. In contrast, resistance in the *hmg1*^pHspA mutant should be resolved as *insA* overexpression would restore the Hmg1-to-InsA balance and, therefore, appropriate negative feedback. Mutant strains overexpressing *insA* were generated through pHspA promoter replacement, as described above, in both the *hmg1*^pHspA and *hmg1*^S305P backgrounds (Fig. 6A). Importantly, *insA* expression levels were unaffected by overexpression of *hmg1* or by expression of the *hmg1*^S305P mutant (Fig. 6B). Additionally, overexpression of *insA* in both the *hmg1*^pHspA (*hmg1*^pHspA/*insA*^pHspA) and *hmg1*^S305P (*hmg1*^S305P/*insA*^pHspA) genetic backgrounds resulted in relative gene expression levels comparable to previously constructed *insA*^pHspA mutant (Fig. 6B). When comparing these new mutants to their parent strains, we noted that the rosuvastatin resistance induced by

overexpression of *hmg1* was not resolved by overexpression of *insA* (Fig. 6C). This again reflects the sterol sensing-independent mechanism of action of the statins. Likewise, terbinafine susceptibility was unaffected by any manipulation (Fig. 6D). In contrast, overexpression of the *insA* gene completely resolved triazole resistance induced by *hmg1* overexpression (Fig. 6E). This result further supports the conclusion InsA plays a conserved role in *A. fumigatus* to mediate negative pathway feedback at the level of Hmg1. Strikingly, overexpression of *insA* in the *hmg1*^S305P SSD mutant was unable to restore to triazole susceptibility (Fig. 6E). Taken together, these findings strongly support the overall hypothesis that triazole stress likely induces accumulation of sterol intermediates that, in turn, promote InsA-mediated negative feedback on Hmg1. The outcome of this feedback is pathway downregulation and reduced growth. Taken further, our results indicate Hmg1 SSD mutations disrupt this normal feedback response to generate triazole resistance.

## Hmg1 SSD mutation partially protects against lethality caused by loss of the C24 sterol methyltransferase, *erg6*

Taken together, our data support the hypothesis that, as in other eukaryotes, sterol intermediates of the *A. fumigatus* ergosterol biosynthesis pathway act as feedback signals to regulate pathway activity (Fig. 7A). Further, triazoles impact a critical step in the pathway to induce heightened accumulation of these sterol intermediates that then induce negative feedback regulation of Hmg1 through the action of InsA (Fig. 7B). Although lanosterol is likely a major sterol intermediate important for this process, triazole-mediated inhibition of the 14α-sterol-demethylase enzyme, Cyp51A/Cyp51B, also generates significant accumulation of the C-24 methylated lanosterol product, eburicol (Fig. 7B). Therefore, either lanosterol or eburicol accumulation could be a major feedback sterol intermediate. To further pinpoint relevant sterol intermediates, we next generated mutant strains that would allow us to specifically

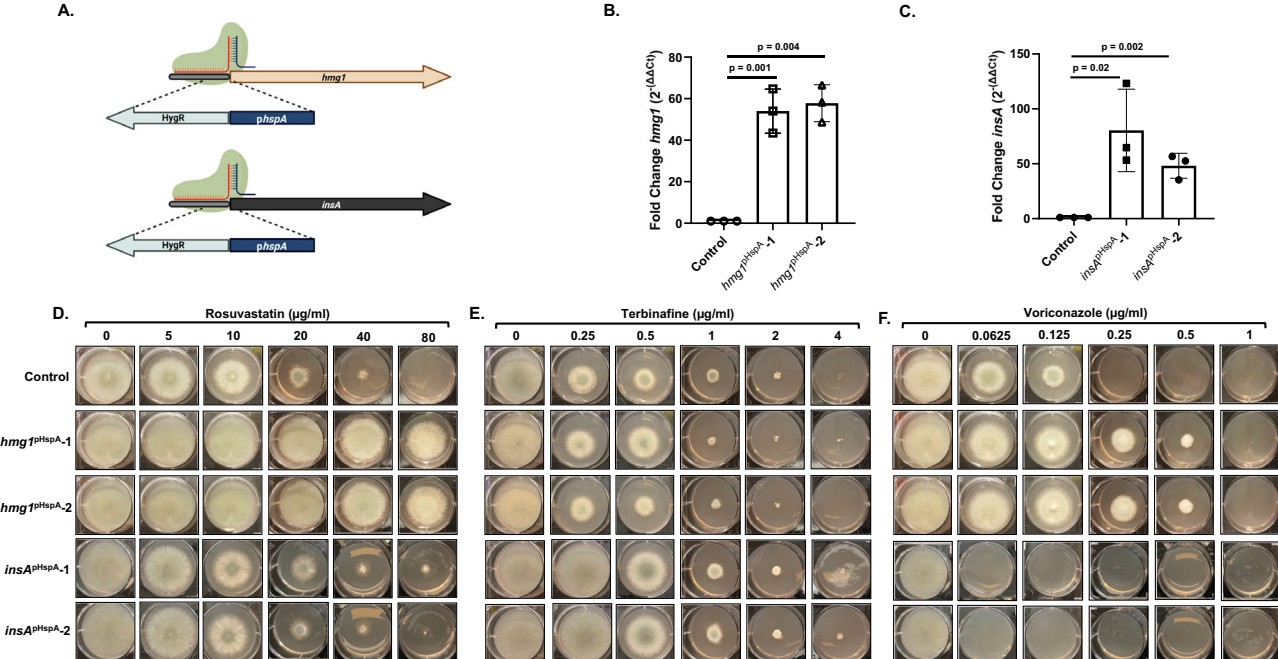

**Fig. 5 | Mutations altering the transcriptional balance of *insA*-to-*hmg1* disrupt normal response to triazole stress. A** Schematic of promoter replacement mutations for *hmg1* and *insA* using the *hspA* promoter (*pHspA*). Illustration prepared using BioRender.com. RT-qPCR analyses to confirm overexpression is shown for *hmg1* (**B**) and *insA* (**C**). Expression levels from two independent mutants for each gene were compared to the parental control. Averaged data represent the mean ± standard deviation. n = three biologically independent samples and statistical analyses were performed by One-Way Anova followed by an unpaired T test

with significance set at 0.05 and degrees of freedom equal to 10 for individual comparisons. Radial growth assays of the parental control, *hmg1* overexpression (*hmg1*pHspA-1 and *hmg1*pHspA-2), and *insA* overexpression (*insA*pHspA-1 and *insA*pHspA-2) mutants to assess susceptibility changes are shown for rosuvastatin (**D**), terbinafine (**E**), and voriconazole (**F**). Conidia (5 ×103) from each strain were spot inoculated onto RPMI media (0.2% glucose, pH 7.0) containing the indicated amount of drug and plates were cultured for 72 hours at 37 °C. Source data are provided as a Source Data file.

induce lanosterol accumulation without inducing eburicol accumulation. To do this, we placed the conserved lanosterol C-24 methyltransferase gene, *erg6*, under the control of a TetOff promoter in both wild type (*hmg1*WT/*erg6*pTetOff) and Hmg1 SSD mutant (*hmg1*S305P/*erg6*pTetOff) genetic backgrounds. This genetic manipulation did not alter basal growth or voriconazole susceptibility in the absence of doxycycline (Supplementary Fig. 3A, B). Importantly, gene expression analyses confirmed that the inclusion of only 0.5 μg/ml doxycycline resulted in *a* ~ 4-fold (log$_2$) decrease in *erg6* expression in both strain backgrounds (Supplementary Fig. 3C). Recent studies from our laboratory have revealed that depletion of *erg6* expression in the *erg6*pTetOff mutant through addition of exogenous doxycycline causes accumulation of lanosterol with no alteration of eburicol[32]. Importantly, we have also uncovered through these studies that the *erg6* gene is essential for viability in *A. fumigatus*. We now reason that, if lanosterol accumulation induces negative feedback of Hmg1, then a portion of the loss-of-viability phenotype in the *erg6*pTetOff should be mevalonate pathway feedback-dependent (Fig. 7C). To see if an intact SSD is required for this phenotype, both the *hmg1*WT/*erg6*pTetOff and *hmg1*S305P/*erg6*pTetOff mutants, along with their parent strains, were cultured in the presence and absence of doxycycline to modulate *erg6* expression. Under all conditions, growth of the parent strain was unaffected (Fig. 7D). As we have previously found, the *hmg1*WT/*erg6*pTetOff mutant rapidly lost viability in doxycycline concentrations as low as 0.25 μg/ml (Fig. 7D). In contrast, the loss-of-viability phenotype upon *erg6* repression was partially ameliorated by mutation of the SSD, as indicated by continued growth of the *hmg1*S305P/*erg6*pTetOff mutant at higher doxycycline concentrations (Fig. 7D). These results, together with data from Figs. 4–6, suggest that lanosterol is likely the important negative feedback sterol for InsA-mediated Hmg1 regulation.

## The conserved ERAD E3 ubiquitin ligase, HrdA, does not contribute to InsA overexpression-mediated triazole hypersusceptibility

As noted above, there are three well-characterized mechanisms of sterol-mediated HMGCR negative feedback (Fig. 8A). These include: i) the INSIG- and E3 ubiquitin ligase (gp78)-dependent accelerated degradation of HMGCR in mammalian cells; ii) the Ins1p-independent, but E3 ubiquitin ligase-dependent, accelerated degradation of Hmg1p in *S. cerevisiae*; and iii) the Ins1-depedent, but Hrd1-independent, phospho-regulation of Hmg1 in *S. pombe*[17]. Interestingly, deletion of the *A. fumigatus* ERAD-associated ubiquitin ligase ortholog, *hrdA*, has previously been shown to generate resistance to voriconazole[33]. When considered with our data supporting an important role for InsA, this finding suggests that a mechanism similar to the mammalian system may exist in *A. fumigatus* wherein triazole-mediated sterol accumulation negatively regulates Hmg1 via InsA/HrdA-controlled accelerated degradation. To test this, we deleted the *hrdA* gene in the wild type and *insA*pHspA genetic backgrounds. We reasoned that, if the conserved HrdA ubiquitin ligase is contributing to the InsA-mediated triazole susceptibility by inducing Hmg1 degradation, then loss of this gene should remove the *insA* overexpression-induced hypersusceptibility to triazoles. Deletion of *hrdA* (Δ*hrdA*) in the wild-type background generated a 4-fold increase in voriconazole resistance (Fig. 8B). This result was similar to that previously reported in another wild-type laboratory strain of *A. fumigatus*[33]. However, deletion of *hrdA* in the *insA* overexpression mutant (*insA*pHspA/Δ*hrdA*) did not cause recovery of wild-type voriconazole susceptibility levels (Fig. 8B). These results suggest that, although *hrdA* deletion generates triazole resistance through an unknown mechanism, this conserved ERAD ubiquitin ligase does not contribute to InsA-mediated triazole responses in *A. fumigatus*. Taken further, these findings support a

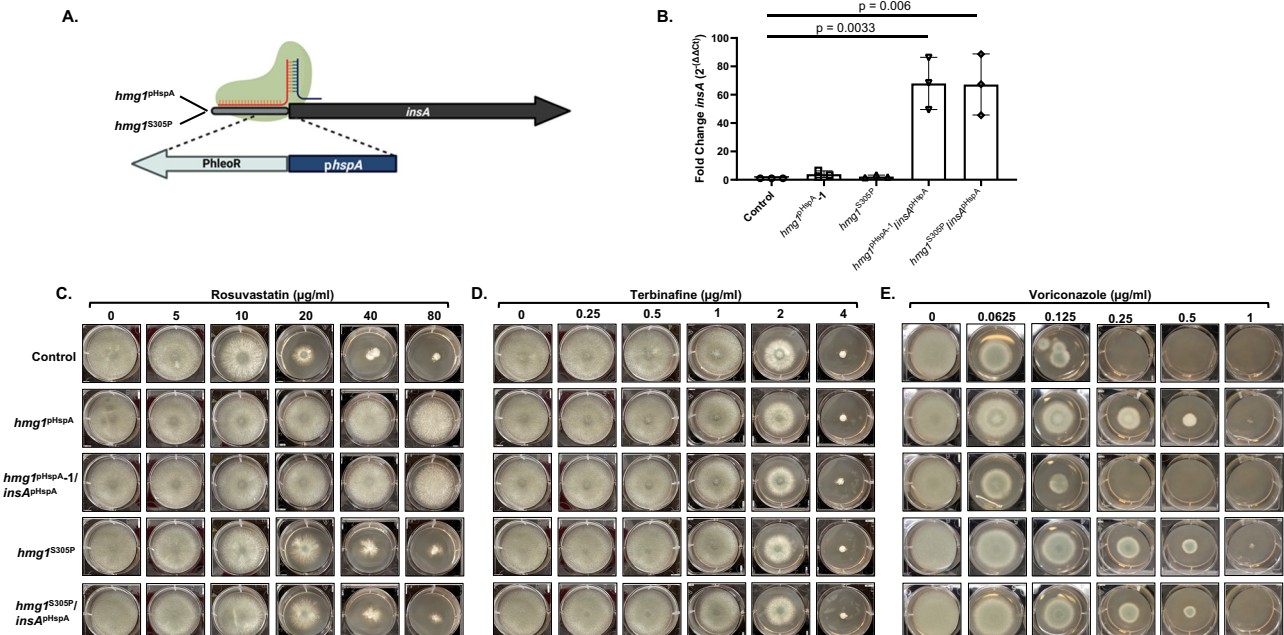

**Fig. 6 | Overexpression of *insA* restores triazole susceptibility to *hmg1* over-expression mutants but does not resolve resistance due to SSD mutation.** **A** Schematic of promoter replacement mutations for *insA* in the *hgm1*[pHspA] and *hmg1*[S305P] genetic backgrounds. Illustration prepared using BioRender.com. **B** RT-qPCR analyses were performed in each background to ensure that the genetic manipulation at the *hmg1* locus of the parental strain did not alter *insA* expression and to confirm that the promoter replacement resulted in overexpression of *insA*. Averaged data represent the mean ± standard deviation. n = three biologically independent samples and statistical analyses were performed by One-Way Anova

followed by an unpaired T test with significance set at 0.05 and degrees of freedom equal to 10 for individual comparisons.Radial growth assays of the parental control, *hmg1* overexpression (*hmg1*[pHspA]), *hmg1*/*insA* double overexpression (*hmg1*[pHspA]/ *insA*[pHspA]), *hmg1* SSD (*hmg1*[S305P]), and *hmg1* SSD/*insA* overexpression (*hmg1*[S305P]/ *insA*[pHspA]) mutants to assess susceptibility changes are shown for rosuvastatin (**C**), terbinafine (**D**), and voriconazole (**E**). Conidia (5 × 103) from each strain were spot inoculated onto RPMI media (0.2% glucose, pH 7.0) containing the indicated amount of drug and plates were cultured for 72 hours at 37 °C. Source data are provided as a Source Data file.

model wherein negative feedback of Hmg1 in *A. fumigatus* might be more similar to the Ins1-dependent, but Hrd1-independent, phospho-regulatory system described for *S. pombe*.

In mammalian cells, INSIG additionally imparts negative regulation on HMGCR by promoting the retention of a transcriptional regulator, SREBP, in the ER under cholesterol-replete conditions[17]. When cholesterol levels are low, INSIG binding is blocked and SREBP is released from the ER, processed for activation, and transported to the nucleus where it induces transcription of cholesterol biosynthesis genes, including HMGCR[34]. Therefore, overexpression of INSIG in mammalian cells both induces accelerated degradation and decreases expression of HMGCR. To see if a similar secondary mechanism for InsA-mediated Hmg1 regulation exists in *A. fumigatus*, we tested for the ability of *insA* overexpression to inhibit function of the SREBP ortholog, SrbA (Supplementary Fig. 2A). To do this, the parent and *insA*[pHspA] mutant strains were cultured in both normoxic and hypoxic conditions on minimal media. Although growth is not impacted under normoxia, in hypoxic conditions strains lacking the *srbA* gene are inviable (Supplementary Fig. 2B)[35]. However, both *insA*[pHspA] mutants were able to grow normally in hypoxia (Supplementary Fig. 2B). These results indicated that the function of SrbA is not inhibited by InsA over-expression and, therefore, mechanistic links between InsA-mediated control of Hmg1 and SrbA are unlikely.

**Triazole stress does not result in reduced abundance or altered sub-cellular localization of the Hmg1 protein**
To further test the hypothesis that the triazole stress-induced negative feedback impacting Hmg1 is independent of protein abundance regulation, we finally sought to measure if Hmg1 SSD mutations affect protein abundance or localization in the presence or absence of drug stress. To do this, strains expressing Hmg1-GFP chimeras were

generated in both the *hmg1*[WT] (*hmg1-gfp*) and *hmg1*[S305P] (*hmg1*[S305P]-*gfp*) genetic backgrounds using CRISPR/Cas9 gene editing for in situ gene modification to allow for expression to be controlled by the native promoter (Supplementary Fig. 4A). We first confirmed that the C-terminal GFP tag did not alter the growth rate (Fig. 9A, B) or the drug susceptibility profiles (Fig. 9C, D) of either strain. Together, these data confirm the Hmg1-GFP chimera is fully functional in each background. Fluorescence microscopy analyses of each new mutant strain revealed GFP localization in a patchwork pattern that coalesced into ring structures evenly dispersed along the length of hyphae (Fig. 9E, F), characteristic of a peri-nuclear endoplasmic reticulum localization in *A. fumigatus*[36]. To see if this localization pattern changed in response to triazole stress, we performed fluorescence microscopy analysis of both strains in the presence of sub-lethal concentrations of voriconazole. We found that, under increasing voriconazole stress, the localization of Hmg1 was largely retained within the peri-nuclear ER in both the *hmg1*[WT]-*gfp* and *hmg1*[S305P]-*gfp* backgrounds with moderate mislocalization to punctae and vesicular structures at 0.0625 μg/ml (Fig. 10A). These results suggest either mislocalization of Hmg1 away from the ER or disorganization of ER membranes in response to triazoles. Although not quantitative, we also noted at the highest concentration of voriconazole tested (0.0625 μg/ml) an apparent increase in fluorescent signal intensity that was not affected by the presence of the S305P SSD mutation (Fig. 10A). These results suggested that, although triazole stress may alter localization of Hmg1 or induce disorganization of ER structure, Hmg1 protein abundance is not reduced.

To assess quantitatively Hmg1 protein abundance, we next adapted a commercially available GFP quantitation assay to measure GFP levels in lysates as a proxy for Hmg1 protein abundance[37]. We first validated that the GFP quantitation assay could detect varying GFP levels in *A. fumigatus* lysates by employing both of our Hmg1-GFP

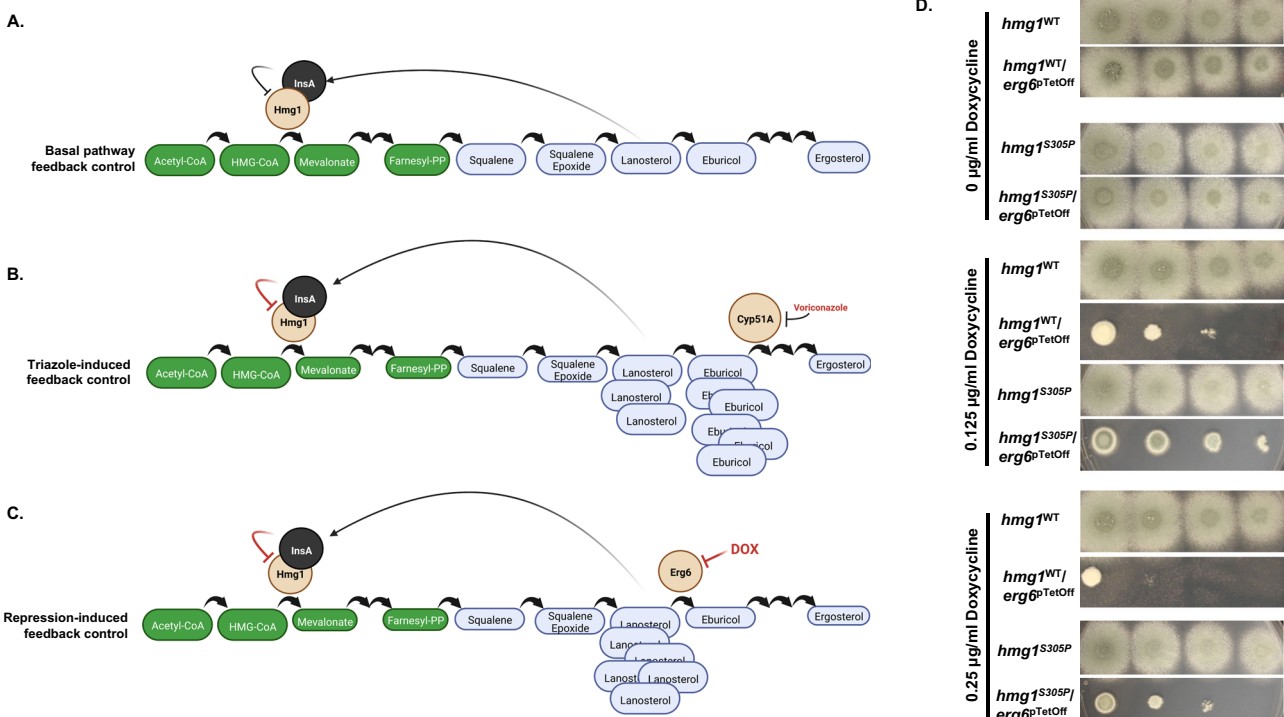

**Fig. 7 | Loss of viability induced by repression of *erg6* is partially blocked by Hmg1 SSD mutation.** Models for sterol intermediate-induced negative feedback of Hmg1 through promotion of InsA binding under normal conditions (**A**), under triazole stress (**B**), and under doxycycline-mediated repression of *erg6* expression in an *erg6*pTetOff mutant (**C**). Illustration prepared using BioRender.com. **D** The parental control strains (*hmg1*WT and *hmg1*S305P) and their *erg6*pTetOff derivatives (*hmg1*WT/ *erg6*pTetOff and *hmg1*S305P/*erg6*pTetOff, respectively) were utilized for spot dilution assays on RPMI media (0.2% glucose, pH 7.0) with the indicated concentrations of doxycycline. Conidial inoculum concentrations for each strain were (from left to right) $5 \times 10^4$, $5 \times 10^3$, $5 \times 10^2$, and $5 \times 10^1$ total conidia. Note the diminished ability of *erg6* repression (i.e., increasing concentrations of exogenous doxycycline) to inhibit growth in the background of an *hmg1* SSD mutant (*hmg1*S305P).

mutant strain, as well as a strain expressing high levels of cytoplasmic GFP from the pOtef promoter[38]. All strains were cultured in the absence of triazole stress, mature mycelia harvest from culture, and lysates were generated from 350 mg of macerated hyphae. GFP abundance was quantitated following the manufacturers protocol. Our results confirmed that the assay was able to reliably detect changes in GFP abundance either through genetic overexpression of GFP or through serial dilution of sample lysates (Supplementary Fig. 4B, C). In addition, our assay validation experiments confirmed that Hmg1-GFP protein abundance was similar between the *hmg1*WT-*gfp* and *hmg1*S305P-*gfp* strains (Supplementary Fig. 4C). To assess the impact of triazole stress, we cultured the Hmg1-GFP mutant strains in the presence of 0, 0.03125, or 0.0625 μg/ml voriconazole, exactly as described for Fig. 10A, and quantitated GFP abundance. As can be seen in Fig. 10B, although a trend towards increased GFP abundance under triazole stress was apparent, no significant differences were noted between the strains under any condition. Therefore, these data suggest that Hmg1-GFP protein abundance is not decreased in response to triazole stress. Taken together, our findings support existence of a triazole-induced Hmg1 negative feedback mechanism that is *insA*-dependent, *hrdA*-independent, and is not mediated through Hmg1 protein degradation.

## Discussion

Since their introduction to clinical practice more than 20 years ago, the mold-active triazole antifungals have been an essential tool in the antifungal armamentarium, heavily relied upon for both the prophylaxis and treatment of numerous filamentous fungal infections, including IA. However, key characteristics of these agents and their activity against filamentous pathogens, such as their fungicidal activity against *Aspergillus* as compared to the fungistatic activity exhibited against yeast such as *Saccharomyces cerevisiae* and *Candida albicans*,

remain poorly understood. Furthermore, the emergence and increasing prevalence of triazole resistance among both agricultural and clinical isolates of pathogenic fungi, including *A. fumigatus*, represents an urgent threat to global public health as recognized by both the United States Center for Disease Control and Prevention and the World Health Organization. Triazole resistance among isolates of *A. fumigatus* is of particular clinical concern given the limited number of mold-active antifungals currently available. While considerable research has been conducted to increase our understanding of the molecular mechanisms contributing to this antifungal resistance, a large proportion of triazole resistance in *A. fumigatus* is not adequately explained by known mechanisms of resistance. This is especially true for resistant isolates in which variations in the known target of the triazoles, *cyp51A*, do not exist.

We previously identified mutations in the *A. fumigatus* HMGCR gene, *hmg1*, as a mechanism of triazole resistance not previously recognized in any other fungal pathogen. Here, through an extensive interrogation of the mechanism by which *hmg1* SSD mutations impart clinical triazole resistance, we have revealed a secondary mechanism of action of the triazoles against *A. fumigatus*: induction of the negative regulation of sterol biosynthesis mediated by the Hmg1 SSD. Although toxic sterol intermediate accumulation has been described for many fungi under triazole stress[2], out findings suggest drug-induced accumulated intermediates inducing negative feedback at rate-limiting steps in the pathway. Importantly, we demonstrate that this role for Hmg1-mediated negative regulation is specific to the triazoles and does not play a role in the antifungal activity of other mevalonate and ergosterol biosynthesis pathway inhibitors such as statins or allylamines. We hypothesize that this is because the triazoles act at a critical step in the ergosterol biosynthesis pathway that, when inhibited, generates accumulation of the negative feedback-inducing sterol

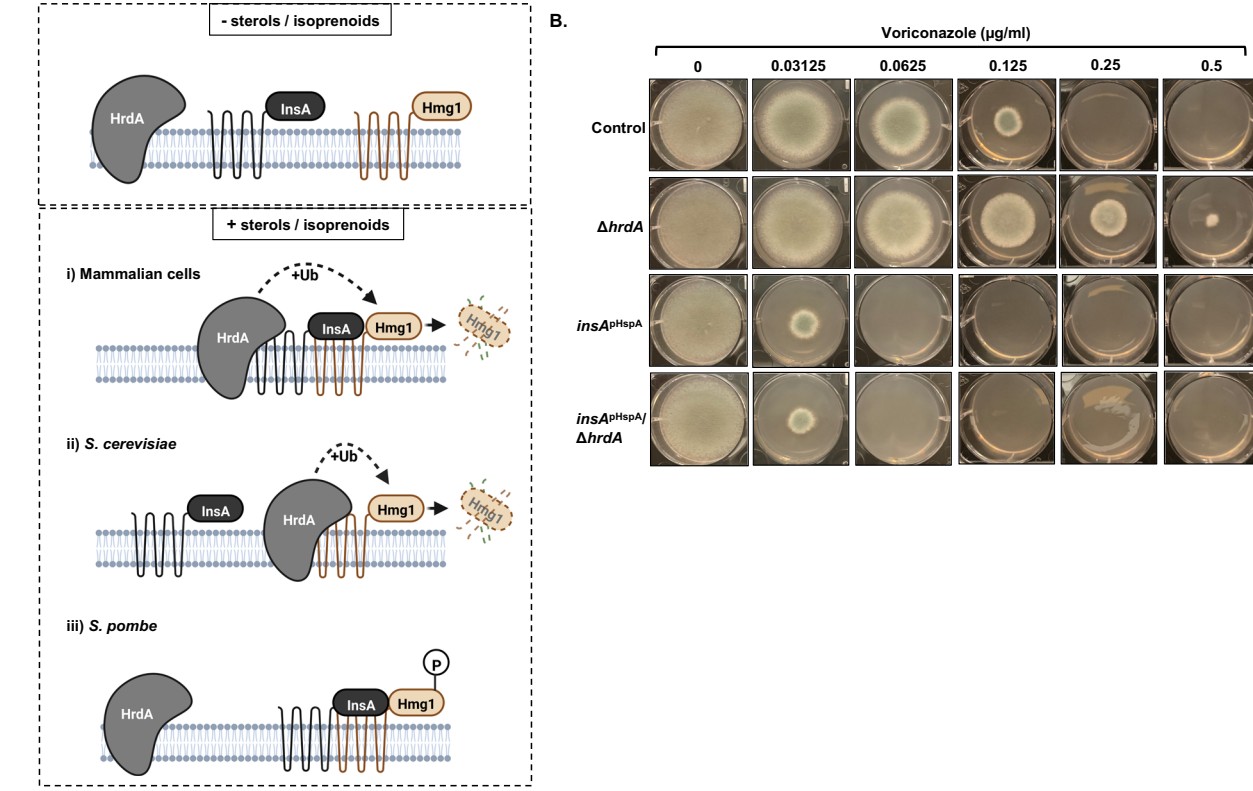

**Fig. 8 | Loss of the conserved ERAD ubiquitin ligase, *hrdA*, does not resolve hypersusceptibility generated by *insA* overexpression. A** Schematic of the potential mechanisms for Hmg1 regulation involving InsA and HrdA based on the known mechanisms in mammalian systems (i), the budding yeast *Saccharomyces cerevisiae* (ii) and the fission yeast *Schizosaccharomyces pombe* (iii). Illustration prepared using BioRender.com. **B** Voriconazole susceptibility assays using the

parental control, *hrdA* deletion strain (Δ*hrdA*), *insA* overexpression strain (*insA*^pHspA), and the *insA* overexpression/*hrdA* deletion double mutant (*insA*^pHspA/ Δ*hrdA*). Conidia (5×10³) from each strain were spot inoculated onto RPMI media (0.2% glucose, pH 7.0) containing the indicated amount of drug and plates were cultured for 72 hours at 37 °C. Source data are provided as a Source Data file.

intermediate, lanosterol. Our findings also highlight how this negative feedback regulation in *A. fumigatus* is similar to the INSIG-regulated mechanisms in other organisms. For example, when HMGCR is over-expressed in mammalian cells, the enzyme is no longer subjected to efficient sterol-mediated negative feedback as the ratio of INSIG-to-HMGCR is artificially altered[20,30]. However, overexpression of either INSIG protein, Insig-1 or Insig-2, restores sterol-mediated negative feedback in mammalian cells with overexpressed HMGCR[30]. Similarly, our mutational analysis suggests negative regulation of the sterol biosynthesis pathway in *A. fumigatus* is dependent on the INSIG-to-HMGCR balance as overexpression of InsA is able to restore suscept-ibility in strains resistant to triazoles due to Hmg1 overexpression. Our findings also suggest the role InsA plays in controlling the response to triazoles is specifically through negative regulation of Hmg1 and not through inhibition of SrbA, the major SREBP involved in triazole-stress response in *A. fumigatus*[39].

SREBPs are transcription factors that, in their inactivate forms, are attached to the ER membrane[40]. Under sterol depleted conditions, SREBPs are transported to the Golgi where they are cleaved for acti-vation and are then shuttled to the nucleus to induce transcription of genes required for sterol biosynthesis[41]. In mammalian cells, INSIG binds to SREBP under sterol-replete conditions through a protein known as SREBP Cleavage Activating Protein (SCAP), leading to retention in the ER and the inability to be activated in the Golgi[17]. The result, then, of sterol intermediate accumulation and/or INSIG over-expression is a double-pronged negative regulation of ergosterol biosynthesis wherein HMGCR activity is downregulated, and SREBP-mediated downstream transcriptional activation of sterol biosynthesis genes is blocked[17,34]. Our data showing that overexpression of InsA

does not phenocopy loss of SrbA, coupled with the lack of a SCAP ortholog in *A. fumigatus*[39], suggests that this mechanistic link does not exist in *A. fumigatus*.

Although the exact mechanism of how InsA imparts negative feedback regulation on Hmg1 in *A. fumigatus* is unclear, we find that it is not through the ERAD-associated ubiquitin ligase ortholog, HrdA. This implies that the mechanism in *A. fumigatus* is highly unlikely to be through the accelerated degradation mechanisms characterized in mammalian cells or in *S. cerevisiae*[17]. However, in *S. pombe*, Ins1 med-iates negative feedback regulation of Hmg1 through phosphorylation of the catalytic domain and a similar mechanism is possible in *A. fumigatus*. In *S. pombe*, alteration of residues in the SSD have been shown to diminish Ins1 interaction with Hmg1[22]. It is feasible then that SSD mutations in *A. fumigatus* Hmg1 could at least partially diminish InsA interactions with Hmg1 in the presence of lanosterol and, there-fore, inhibit negative regulation. As we note that they commonly occur in isolates that also bear Cyp51A mutations, it is interesting to spec-ulate Hmg1 SSD mutations could initially arise to generate low-level resistance and subsequently potentiate target gene mutation for selection of high-level resistance. In fact, we, and others, have descri-bed high-level triazole resistance in clinical isolates carrying *hmg1* and *cyp51A* combination mutations[42]. Future work will focus on decipher-ing the exact mechanism through which triazoles impart negative feedback and if their presence potentiates development of high-level resistance. Regardless of the exact feedback mechanism, this work highlights the need for future studies to consider the induction of negative feedback on Hmg1 as an important component when working to improve triazole activity or develop improved ergosterol bio-synthesis inhibitors.

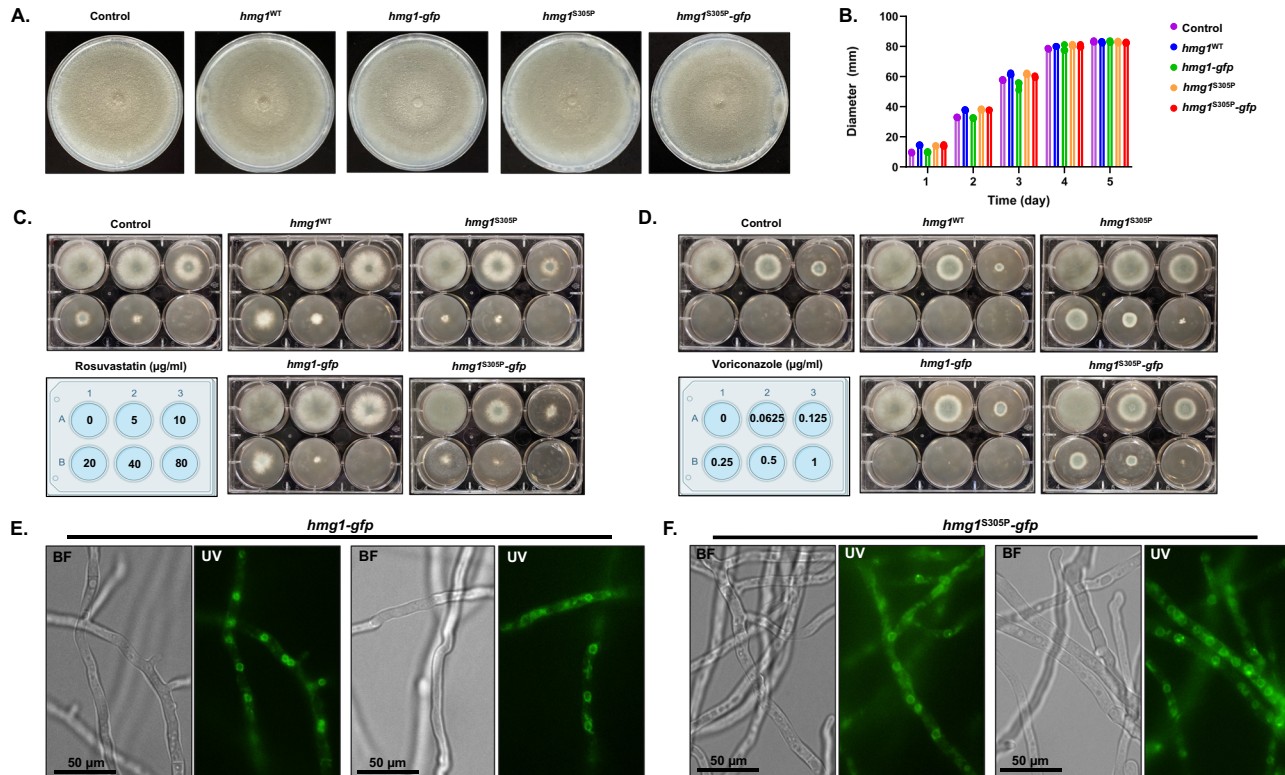

**Fig. 9 | Hmg1 SSD mutations do not alter localization of Hmg1 to the *A. fumigatus* endoplasmic reticulum. A** Colony morphology comparison of the control, Hmg1 parental (*hmg1*^WT^ and *hmg1*^S305P^), and Hmg1-GFP strains (*hmg1*^WT^-*gfp* and *hmg1*^S305P^-*gfp*) after 5 days growth on minimal media. **B** Quantitation of colony diameter for the indicated strains at days 1, 2, 3, 4, and 5 post-inoculation. Assays were completed in biological triplicate. **C** Rosuvastatin and **D** voriconazole susceptibility assays for the control, Hmg1 parental, and Hmg1-GFP strains. Assays were completed as described in Figure X. Drug concentrations for each well are noted in the included plate diagrams. **E** Hmg1^WT^-GFP and **F** Hmg1^S305P^-GFP localization analyzed by epifluorescence microscopy. Conidia from each strain were inoculated into minimal media, incubated for 12 hrs at 37 °C, and then mounted for microcopy. Images were captured using an exposure time of 600 ms. Note localization to regularly spaced ring structures indicative of *A. fumigatus* peri-nuclear endoplasmic reticulum. BF = brightfield; UV = ultraviolet. Microscopy experiments were completed three times independently with similar results. Source data are provided as a Source Data file.

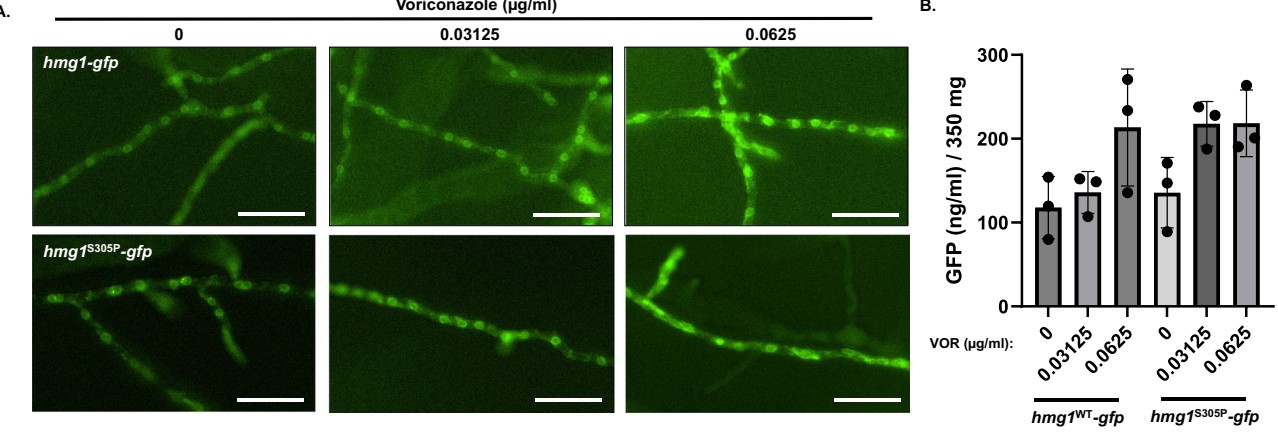

**Fig. 10 | Triazole stress does not reduce Hmg1 protein abundance.**
**A** Localization analyses of Hmg1-GFP in response to sub-lethal triazole stress in the *hmg1*^WT^-*gfp* and *hmg1*^S305P^-*gfp* strains. All images were captured with an exposure time of 600 ms. Note apparent increase in intensity and mislocalization of the GFP fluorescent signal at 0.0625 μg/ml voriconazole. Scale bar = 50 μm. Microscopy experiments were completed three times independently with similar results.
**B** Quantitation of GFP in lysates of the *hmg1*^WT^-*gfp* and *hmg1*^S305P^-*gfp* strains cultured for 12 hrs in the presence of 0, 0.03125, or 0.0625 μg/ml voriconazole. Data are presented as the mean ± standard deviation. Assays were performed in biological triplicate and statistical analyses completed using one-way ANOVA and Tukey's Test post-hoc. Despite a trend for increased GFP abundance in the presence of voriconazole, no significant differences were noted among the groups. Source data are provided as a Source Data file.

## Methods

### Ethics Statement

All research was performed in compliance with all ethical regulations under protocol 22-1019 approved by the Institutional Biosafety Committee at the University of Tennessee Health Science Center. All clinical isolates used in this study were obtained from the Fungus Testing Laboratory at the University of Texas Health Science Center in San Antonio, Texas or from the US Centers for Disease Control in

Atlanta, Georgia. All isolates were obtained with no patient data and therefore consent was not required.

## Strains and growth conditions used in this study

The triazole-susceptible control strain used for all genetic manipulations in this study was Δ*akuB-pyrG⁺*, a uridine/uracil prototrophic strain previously generated in the Δ*akuB*^KU80 genetic background[43]. All clinical isolates used in this study are found in Supplementary Data 1. Strains and clinical isolates used in this study were maintained on glucose minimal medium (GMM) agar at 37 °C[44]. All conidia were harvested in sterile water from 3-day-old growth plates, and conidia were enumerated visually using a hemocytometer before use in growth assays. Our collection included newly acquired multitriazole-resistant clinical isolates originating from the United States that were obtained from the Fungus Testing Laboratory at the University of Texas in San Antonio and from the Centers for Disease Control and Prevention (Supplementary Data 1). Whole-genome sequencing and triazole susceptibility profiles were completed as part of this study (see below). Sequences from additional isolates were selected from 8 total external NCBI bioprojects based on the availability of WGS Illumina reads and if there were EUCAST or CLSI measured MIC values for at least one of the four triazoles of interest (Voriconazole, Posaconazole, Isavuconazole, and Itraconazole). Isolates for analysis were taken from the following NCBI bioprojects: PRJEB27135, PRJNA673120, PRJEB8623, PRJNA67101, PRJDB7240, PRJNA592352, and PRJNA638646.

## Whole-genome sequencing, variant analysis, and phylogenetics

For each isolate without publicly available genomic sequence, genomic DNA was extracted using the Qiagen DNeasy Plant Mini Kit as previously described[45]. DNA concentrations were quantified using both the Qubit fluorometer and the NanoDrop spectrophotometer using the manufacturers' protocols. Whole-genome libraries were prepared and sequenced by the Hartwell Center at St. Jude Children's Research Hospital. Genomic DNA was quantified using the Quant-iT PicoGreen ds DNA assay (ThermoFisher). Genomic DNA was sheared on an LE220 ultrasonicator (Covaris). Libraries were prepared from sheared DNA with the HyperPrep Library Preparation Kit (Roche PN 07962363001). Libraries were analyzed for insert size distribution using the 2100 BioAnalyzer High Sensitivity kit (Agilent), 4200 TapeStation D1000 ScreenTape assay (Agilent), or 5300 Fragment Analyzer NGS fragment kit (Agilent). Libraries were quantified using the Quant-iT PicoGreen ds DNA assay (ThermoFisher) or by low-pass sequencing with a MiSeq nano kit (Illumina). Paired end 150 cycle sequencing was performed on a NovaSeq 6000 (Illumina). The *A. fumigatus* assembly strain AF293 (GCA_000002655.1) was used as the reference genome for Illumina sequences for isolates to perform variant calling. All isolate sequences were processed on Terra.bio. Isolate-paired FASTQs were processed into unmapped BAM files using the Terra workflow paired-fastq-to-unmapped-bam (https://portal.firecloud.org/?return=terra#methods/gatk/paired-fastq-to-unmapped-bam/10) to run the GATK command FastqToSam. The output unmapped BAM files were then run through the fungal-variant-call-gatk4 workflow (https://github.com/broadinstitute/fungal-wdl/tree/master/gatk4), which implements the GATK HaplotypeCaller (v. 4.1.8.1) for both SNPs and indels[46]. Next, the per-sample GVCF files were combined and genotyped with CombineGVCFs and GenotypeGVCFs. Selected variants were filtered with VariantFiltration using "QD < 2.0 || FS > 60.0 || MQ < 40.0." Genotypes were filtered with a script in this workflow, requiring a minimum genotype quality of <50, percent alternate allele of <0.8, or depth of <10. The final variant calling format (VCF) file was annotated and given functional predictions using SnpEff (v. 4.3-t), and also filtered for variants with a PASS flag using vcftools (v. 0.1.15)[47,48]. To reduce the number of false-positive identified variants, filtering of spanning deletions was also implemented by searching for variants with an alternative allele containing an asterisk. To infer a phylogeny to represent the relationships between isolates, the VCF file was converted into FASTA format using a python script (https://github.com/broadinstitute/broad-fungalgroup/blob/master/scripts/SNPs/vcfSnpsToFasta.py). Maximum likelihood phylogenies were built using RAxML (v. 7.7.8), with the GTRCAT nucleotide substitution model and 1000 bootstrap replicates[49].

## RNA sequencing and differential gene expression analysis

For RNA sequencing and DGE analysis, strains were cultured in RPMI (0.2% glucose, pH 7.0) in the presence of 0.5 X MIC voriconazole or vehicle control for 48 hours in an orbital shaker at 250 rpm and 37 °C. Hyphae were harvested by filtration, washed with sterile water and flash frozen in liquid nitrogen. Total RNA was extracted using TRIzol and the RNeasy Mini Kit (Qiagen), as previously described[50]. All RNA-seq libraries (strand-specific, paired end) were prepared with the TruSeq RNA Sample Prep kit (Illumina). The total RNA samples were subject to poly(A) enrichment as part of the TruSeq protocol. In total, 75 nt of sequence was determined from both ends of each cDNA fragment using the HiSeq platform (Illumina) as per the manufacturer's protocol. Reads were aligned to the *A. fumigatus* reference genome (Af293) using HISAT2[51]. Aligned reads were then used to generate read counts for each gene and the DE-seq package from Bioconductor was used for statistical analysis of differential gene expression[52]. A gene was considered differentially expressed if the FDR for differential expression was less than 0.05 and the absolute log2 fold-change (LFC) was greater than or equal to one. The RNA-seq analysis was performed in biological triplicate. For GO analyses of significantly differentially expressed genes, statistical analysis was performed by Fishers exact test for determination of p-value and Benjamini-Hochberg procedure for adjusted p-value.

## Comprehensive sterol profiling

Sterol profiling was completed as previously described, with minor modifications[8]. Briefly, fresh conidial suspensions of each strain to be studied were prepared in saline with Tween 80 from Aspergillus minimal medium agar plates. Conidia were grown in RPMI (0.2% glucose, pH 7.0) in the presence of 0.5 X MIC voriconazole or vehicle control for 48 hours in an orbital shaker at 180 rpm and 35 °C. Cells were then flash-frozen using liquid nitrogen, dry weights were obtained, internal standard of cholesterol added to the cells, sample contents saponified with alcoholic KOH and hexane was utilized to extract nonsaponifiable lipids. A vacuum centrifuge was used to dry samples, prior to derivatization by the addition of 200 μl of anhydrous pyridine, 100 μl of N,O-bis(trimethylsilyl)trifluoroacetamide (BSTFA)−10% trimethylsilyl (TMS), and 2 hours of heating at 80 °C. Gas chromatography-mass spectrometry (GC-MS) (a DB-5ms column (J & W Scientific, cat. No. 122-5532UI), Thermo 1300 GC coupled to a Thermo ISQ mass spectrometer (Thermo) was then used to analyze and identify TMS-derivatized sterols. Splitless injection of 1 μL of sample was performed with an initial oven temperature of 70 °C for 4 min, ramping 25 °C/ min to a final temp of 280 °C and held for 25 min. The EI ion source was held at 280 °C and operated at 70 eV. Data was acquired in TIC mode (50-600 m/z) with a solvent delay of 14 min. Known standards were referenced for fragmentation spectra and retention times. Sterol profiles for each isolate were then created using Xcalibur software (Thermo Scientific) to analyze GC-MS data[53]. Statistical analysis of both sterol profiles and total ergosterol per dry weight was performed using GraphPad Prism 7. In all cases, 6 independent biological replicates were measured and included in analysis, and two-tailed unpaired t tests were performed in Prism 8 for Mac OS by GraphPad Software Inc. with significance set at 0.05 and degrees of freedom equal to 10.

## Genetic manipulations for construction of mutant strains

Generation of strains harboring *hmg1* mutant alleles and *hspA* promoter replacement mutations were performed as previously described[8]. All primers and crRNA sequences utilized for the manipulation described below can be found in Supplementary Data 1. In brief, for *hmg1* allele replacement mutations, two-component repair templates consisting of a split hygromycin resistance marker (*hphR*) and *hmg1* alleles of interest were prepared by PCR. The *hmg1* alleles of interest included the open reading frame and approximately 500 downstream bases and were amplified by PCR from DI-20-100 (E105K), DI-20-118 (D242H), DI-20-135 (W272L), DI-20-110 (C402R), DI-20-84 (V403D), DI15-96 (G466V/S541G), DI-20-129 (G483R), DI-20-116 (V995I), DI-20-89 (V995I/W272L), and DI-15-97 (V995I/S305P) and Δ*akuB-pyrG*⁺ (Control) using a 3′ primer which introduced the terminal 80 bases with homology to the 3′ end of the *hphR* hygromycin B resistance gene open reading frame. The sole exception to this approach was employed for the L558R mutation, wherein a synthetic gene construct was custom designed (Integrated DNA Technologies) and then utilized for PCR-based repair template generation rather than starting with genomic DNA. A partial hygromycin B resistance cassette, including the *gdpA* promoter and a truncated *hphR* gene lacking the terminal 40 bases, was then amplified by PCR from the pUCGH plasmid using primers that introduced approximately 70 bases of homology with the downstream region of *hmg1*. Cas9-RNP complexes targeting protospacer/PAM sequences immediately upstream and approximately 500 bases downstream of the open reading frame of *hmg1* were assembled as previously described[8]. Custom, target-specific crRNA guide sequences are listed in Supplementary Data 1. For overexpression of *hmg1* and *insA*, promoter replacement transformation repair templates were generated by PCR using primers which amplified both the hygromycin resistance cassette and the *hspA* promoter from the plasmid pJMR2[8], while also introducing 40 bases of homology targeting sequences immediately upstream and downstream of the start codon for the respective gene of interest. Cas9-RNP complexes targeting sequences immediately upstream of the open reading frame of each gene of interest were assembled as previously described[26]. To construct strains harboring doxycycline-regulatable *erg6*, we first built a plasmid carrying a phleomycin resistance cassette (*phleoR*) and the TetOff promoter system. To do this, we first digested the TetOff plasmid pSK606[54] with *BglII* and *KpnI* to remove the pyrithiamine resistance cassette (*ptrA*). A phleomycin resistance cassette was PCR-amplified from plasmid pAGRP[44] to contain *BglII* and *KpnI* restriction sites at the 5′ and 3′ ends, respectively, and subsequently sub-cloned into the digested pSK606 plasmid. The resulting plasmid was named pSK606-phleo. PCR primers were designed to amplify the *phleoR*-pTetOff cassette from pSK606-phleo for a repair template to perform CRISPR/Cas9-based gene editing, as described[26]. Primers were designed to incorporate 40 bp of homology upstream and downstream of a PAM site selected immediately upstream of the predicted *erg6* start codon. Transformations of the resulting *phleoR*-pTetoff repair template were performed in the *hmg1*^WT and *hmg1*^S305P genetic backgrounds. Proper integration of the *phleoR*-pTetOff construct upstream of *erg6* was confirmed by PCR, Sanger sequencing, and in vitro growth screens in the presence of doxycycline. For the deletion of *hrdA*, PAM sites and protospacer sequences were first selected upstream of the predicted start codon and downstream of the predicted stop codon, as previously described[26]. Primers were designed to amplify a hygromycin resistance (*hphR*) cassette from plasmid pUCGH[55] while incorporating 40 bp of homology to either side of the *hrdA* 5′ and 3′ PAM sites. The resulting *hphR* repair template was transformed into the control and *insA*^pHspA genetic backgrounds. Complete deletion of *hrdA* was confirmed by PCR. All genetic manipulations were confirmed by Sanger sequencing.

Generation of C-terminal GFP fusion mutants was performed using CRISPR/Cas9 gene-editing to in situ tag the *hmg1*^WT and

*hmg1*^S305P genes in each respective genetic background. In brief, a suitable PAM site and protospacer were selected at the 3′ end of the putative *hmg1* coding sequence and a custom gRNA was designed and constructed, as previously described[26]. Transformations were carried out using a repair template consisting of the eGFP sequence upstream of a phleomycin resistance cassette (eGFP-*bleoR*), flanked by 40-basepair regions of homology targeting the chosen PAM. Proper integration of the eGFP-*bleoR* cassette was screened by diagnostic PCR and in-frame fusions were confirmed by Sanger sequencing.

## Assessment of gene expression by RT-qPCR

For assessment of gene expression after promoter replacement, conidia from each strain were cultured for 24 hours in RPMI medium (0.2% glucose, pH 7.0) at 37 °C on an orbital shaker at 250 rpm. After incubation, hyphae were harvested from culture by filtration and washed with sterile water. Total RNA was then extracted following liquid nitrogen freezing and pulverization as previously described[8]. The iScript cDNA synthesis kit (BioRad) was utilized to synthesize cDNA and iQ SYBR Supermix (BioRad) was utilized to amplify *A. fumigatus hmg1*, *insA*, and *tubA* from cDNA by PCR per the manufacturer's instructions. Supplementary Data 1 lists the gene-specific primers used for PCR. The qPCR reactions were performed on the CFX96 real-time PCR system (Bio-Rad) and the dissociation curve and threshold cycle (cT) values were determined using the CFX Maestro software package (bioRad). Changes in gene expression among isolates were then calculated using the $2^{-(\Delta\Delta CT)}$ method. All experiments were performed in triplicate from biological triplicates. As previously described, ΔCT values were used to calculate the standard error[50]. Statistical analysis was performed using unpaired, two-tailed, Student's t test in Prism 8 by GraphPad Software Inc. with significance set at 0.05 and degrees of freedom equal to 10.

## GFP quantitation assay

To quantitate GFP in *A. fumigatus* lysates, conidia ($10^5$) from each strain were inoculated into 100 ml of Glucose Minimal Media (GMM) broth in the presence of either vehicle (0), 0.03125, or 0.625 μg/ml voriconazole and cultured for 16 h at 37 °C. GMM was utilized rather than RPMI to avoid fluorescence readout interference associated with pH-indicator dye in commercially prepared RPMI. After incubation, hyphae were harvested from culture, dried and crushed under liquid nitrogen to form a fine powder. To generate lysates from the fungal powder and to assay GFP abundance, the GFP Quantitation Kit (Cell Biolabs) was utilized, following the manufacturer's protocol, with minor modifications. Briefly, 350 mg of fungal powder from each strain was resuspended in 4 ml of water to generate a crude lysate to which 1 ml of the manufacturer Quantitation Buffer was added. The samples were then vortexed for 30 s followed by centrifugation for 8 min at 3500 rpm to generate a cleared lysate. After centrifugation, 200 μl of cleared lysate from each sample was loaded into a black-masked 96-well plate and GFP intensity recorded on a fluorescent plate reader using GFP filter settings. For conversion of fluorescence intensity readings to GFP abundance, a standard curve was generated using the GFP standard provided by the manufacturer and following the manufacturer's instructions. All measurements were performed with, at least, three technical replicates and three biological replicates for each sample. For initial validation of the ability of the GFP Quantitation Kit to reliably measure GFP abundance in *A. fumigatus* lysates, a mutant strain expressing high abundance cytosolic GFP from a strong constitutive promoter (pOtef)[38] was utilized coupled with 1:2 serial dilutions of the cleared lysate.

## Fluorescence microscopy

To assess localization of Hmg1-GFP chimeric proteins, conidia ($10^5$) from each GFP-fusion mutant were inoculated into GMM broth,

aliquoted into individual wells of a 24-well glass-bottomed petri dish and incubated for 16 hrs at 37 °C. Cultures were performed in the presence of vehicle (0), 0.03125, or 0.625 µg/ml voriconazole. After incubation, images were captured using a Nikon inverted fluorescence microscope equipped with a GFP filter and a standard exposure time of 600 ms. Representative images from each culture are shown and all assays were performed in biological triplicate.

## Clinical antifungal susceptibility testing

As previously described[8], susceptibilities for voriconazole, isavuconazole, itraconazole, and posaconazole were determined for all strains in accordance with CLSI M38-A2 methodology utilizing broth micro-dilution in RPMI[56]. Each antifungal was obtained from the appropriate manufacturer. All agents were suspended in dimethyl sulfoxide (DMSO) for preparation of stock solutions.

## Statistics and reproducibility

No statistical tests were utilized to pre-determine sample size. As it is the minimum number of replicates required for inferential analysis, at least three biological replicates were utilized for all experiments. No data were excluded from analyses, the experiments were not randomized, and investigators were not blinded to allocation during experiments of outcomes assessment. Statistical analyses were performed using GraphPad Prism 10.0.0 for Windows (GraphPad Software, San Diego, CA, USA). Specific tests used to determine statistical analyses are noted in each figure legend. $p$ values are depicted, with a value of $p < 0.05$ considered significant.

## Reporting summary

Further information on research design is available in the Nature Portfolio Reporting Summary linked to this article.

## Data availability

Whole-genome and transcriptome sequencing data files for the *A. fumigatus* isolates sequenced as part of this study have been deposited in NCBI SRA under the accession numbers PRJNA985736 and PRJNA991520, respectively. Source data are provided as a Source Data file. Source data are provided with this paper.

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

## Acknowledgements

This work was supported by the National Institutes of Health (NIH)/ National Institute of Allergy and Infectious Diseases (NIAID) grant R01 AI143197 (J.R.F./P.D.R.) and grant U19 AI110820 (V.M.B.). We are grateful for the SJCRH Hartwell Center (supported in part by the ALSAC and the National Cancer Institute grant P30 CA021765) for their expertize in generating the whole genome sequencing data. The authors would like to thank Nathan P. Wiederhold, PharmD and Shawn R. Lockhart, PhD at the Fungus Testing Laboratory in San Antonio and at the Centers for Disease Control and Prevention, respectively, for providing isolates used in this study. The authors would also like to thank Sven Krappman for the kind gift of the pSK606 plasmid. Portions of figures were created with BioRender.com.

## Author contributions

Conceptualization and planning: J.R.F., P.D.R., and J.M.R.; Experimentation: J.R.F., W.G., J.X., A.M-V., X.G., A.V.N., H.I.T., and J.E.P.; Data analysis: J.R.F, J.E.P., C.A.C, J.M.R., J.X., A.M-V., H.I.T, W.G., A.C.O.S., H.M.S., V.M.B., A.C.S., and C.M.; Manuscript and figures preparation: J.R.F, J.E.P., C.A.C, J.M.R., J.X., A.M-V., H.I.T, W.G., A.C.O.S., H.M.S., V.M.B., A.C.S., and C.M.

## Competing interests

The authors declare no competing interests.

## Additional information

[1]Department of Pharmacy and Pharmaceutical Sciences, St. Jude Children's Research Hospital, Memphis, TN, USA. [2]Graduate Program in Pharmaceutical Sciences, College of Pharmacy, University of Tennessee Health Science Center, Memphis, TN, USA. [3]Department of Clinical Pharmacy and Translational Science, College of Pharmacy, University of Tennessee Health Science Center, Memphis, TN, USA. [4]Integrated Program in Biomedical Sciences, College of Graduate Health Sciences, University of Tennessee Health Science Center, Memphis, TN, USA. [5]Department of Microbiology, Immunology, and Biochemistry, College of Medicine, University of Tennessee Health Science Center, Memphis, TN, USA. [6]Institute of Genome Sciences, University of Maryland School of Medicine, Baltimore, MD, USA. [7]Department of Microbiology and Immunology, University of Maryland School of Medicine, Baltimore, MD, USA. [8]Molecular Biosciences Division, School of Biosciences, Cardiff University, Cardiff, Wales, UK. [9]Institute of Life Science, Swansea University Medical School, Swansea, Wales, UK. [10]Infectious Diseases and Microbiome Program, Broad Institute of MIT and Harvard, Cambridge, MA, USA. [11]These authors contributed equally: P. David Rogers, Jarrod R. Fortwendel. ✉e-mail: jfortwen@uthsc.edu

