## [Peer Review File · Nature Communications]

A secondary mechanism of action for triazole antifungals in *Aspergillus fumigatus* mediated by *hmg1*REVIEWER COMMENTS

Reviewer #2 (Remarks to the Author):

Triazole antifungals targeting ergosterol biosynthesis are key for therapy of fungal infections including invasive aspergillosis. Recently, Jarrod R. Fortwendel's group identified a novel cause for resistance against triazoles in *Aspergillus fumigatus*, which is the major cause of invasive aspergillosis, based on mutations within the gene encoding HMG-CoA reductase/HMGCR (Reference 10). Here, the same group (Rybak et al.) clarified the underlying molecular mechanism via elegant genetic studies. They show that the identified mutations in HMGCR disrupt sterol sensing, actually sensing of ergosterol pathway intermediates (most likely lanosterol) that accumulate during triazole inhibition of Cyp51A. Sterol sensing by HMGCR together with insig has previously been reported in *Saccharomyces cerevisiae*, *Schizosaccharomyces pombe* and mammals. However, the exact mechanisms show significant differences in these organisms, as described in the manuscript. Therefore, the clarification of this issue in the opportunistic pathogen *A. fumigatus* is of high clinical relevance.

Specific comments

- 1) In the Abstract and the main text (e.g., line 71) describe the action of azoles exclusively as "ergosterol depleting" and consequently their identified mechanism several times in the manuscript as "novel secondary mechanism" (e.g., line 43, 50, 135, etc). However, it has been shown in several fungal species that azole treatment results not only in depletion of ergosterol but concomitantly to accumulation of toxic pathway intermediates that disrupt the cell membrane (PMID: 25320648). This has been neglected in this manuscript and should be complemented.
- 2) Recent studies revealed that there are two to three primary populations/clades in *A. fumigatus* (PMID: 36395320, PMID: 35469019). It would be interesting to know if the azole resistance causing mutations within HMGCR are population/clade specific or not.
- 3) Line 185: Which mutant strains? I assume CEA10 strains with introduced mutations.
- 4) Line 192: "upregulated in common" compared to what?; this page: wouldn't it be interesting to comment also the downregulated genes?
- 5) Line 276: The fact that overexpression of *hmg1* increases resistance to triazoles but not terbinafine has been reported previously (PMID: 36350143).
- 6) Figure 7D: the strain *hmg1(WT)erg6(pTetOff)* shows decreased growth without doxycycline (most upper row) compared to *hmg1(WT)*, which might be caused by overexpression of *erg6* as this is driven by unrepressed pTetOff. If this holds true, this should result in lanosterol depletion. It might be interesting to assay triazole susceptibility of strain *hmg1(WT)erg6(pTetOff)* compared to *hmg1(WT)*.
- 7) In experiments using the pTetOff promoter, the effect of doxycycline on expression of the respective gene should be shown, at least exemplary.

Reviewer #3 (Remarks to the Author):

Infections due to the fungal mold *Aspergillus fumigatus* cause a spectrum of respiratory diseases ranging from asthma to chronic and life-threatening invasive pulmonary aspergillosis. In the latter case, antifungal therapy is critical for clinical response and highly active triazole antifungal agents are recommended as first-line therapy. Resistance to azole antifungal agents can occur following prolonged and repeated clinical exposure but more recently have been identified with environmentally-derived resistant strains caused by extensive use of azoles in agriculture. Such resistant isolates are associated with clinical failures following initiation of therapy. As azoles

interfere with ergosterol biosynthesis, most resistance is known to be associated with specific mutations within Cyp51A, which encodes the azole drug target 14C-demethylase enzyme, a key enzyme in the biosynthetic pathway. However, a subset of resistant isolates is not associated with Cyp51A mutations and represent a different molecular bypass pathway. Such bypass/feedback pathways involving HMG-CoA reductase (HMGCR) have previously been identified by the authors and others in the field. They had proposed that mutations in the sterol sensing domain (SSD) might result in a loss of sterol sensing, which might block pathway feedback regulation as sterol intermediates are produced. The current study was undertaken to perform a deeper dive and more comprehensive analysis of inhibition of ergosterol biosynthesis by triazole antifungal agents and investigate how specific conserved sterol regulatory factors centered on HMG-CoA contribute to residual azole resistance. Overall, the work in this study is comprehensive and well executed. It brings to life important insights of sterol regulation, which highlights important similarities and differences described for cholesterol biosynthesis in humans. There are still gaps in fully understanding signaling and how resistance occurs, but this mechanism is a step forward for the field as it reveals a another, albeit secondary, resistance mechanism for azoles. This work is most relevant to specialists in medical mycology given that this new mechanistic insight is limited in induction by certain highly active triazole antifungals.

The overall study provides a firm footing for this novel mechanism of resistance, which appears specific for highly active triazole antifungals and not other sterol biosynthesis inhibitors.

Specific Comments

The Introduction wanders at times and should be condensed. At times, it appears more like a Discussion than an Introduction. These wanderings detract from the impending objective, especially for non-specialists.

Lines 142. Can you clarify the origin of the global clinical isolates obtained. It is stated in the Methods that isolates were obtained from the Fungus Testing Lab and represented newly acquired US isolates. This might be somewhat problematic as US isolates are rarely drug resistant. Please clarify origin.

Line 173. The MIC ranges are not very much higher than susceptible strains (MIC 2-8 fold). How do the authors know that these isolates are drug resistant and not drug tolerant? The former would be untreated or poorly treated in animal models and would be expected to induce a higher probability for clinically refractive responses. The authors might want to consider that while this new mechanism may elevate MIC, it might induce resistance in some cases but may be more important for tolerance development. Such tolerance would predispose such isolates downstream for generating Cyp51A mutations. Have the authors observed SSD mutations in isolates with Cyp51A mutations?

Reviewer #4 (Remarks to the Author):

Triazoles, which inhibits ergosterol biosynthesis, are the frontline antifungals for treating life-threatening human fungal infections. However, many triazoles resistant clinical isolates have been isolated in recent years. The main mechanism of azole resistance is believed to be due to mutations in the triazole target gene *cyp51A*. However, recent studies from this and other groups have found many triazoles resistant isolates lacking *cyp51A* mutation and identified mutations in other genes including *hmg1*, which encodes the rate-limiting enzyme HMG-CoA reductase in the mevalonate biosynthesis necessary for ergosterol biosynthesis. It was also found that the *hmg1* mutations are enriched in the sterol sensing domain (SSD) of Hmg1, which is important for its enzymatic activity, and proposed that the sterol sensing function and negative regulation of Hmg1 affect triazoles resistance. This manuscript now provides experimental evidence to support the proposed mechanism. Specifically, the manuscript (1) expanded the mutation analysis on a larger collection of drug resistant clinical isolates showing that Hmg1 SSD mutations are common, (2) demonstrated that SSD mutations alters *erg* gene expression and sterol accumulation, (3) showed

that Hmg1 and its potential regulator InsA levels are important for triazoles resistance, (4) showed that Lanosterol is likely the sterol sensed by the SSD for Hmg1 regulation, and (5) showed that the InsA-mediated regulation of Hmg1 is independent of the function of the conserved ERAD E3 ubiquitin ligase and the SrbA transcription factor, the orthologs of which have been shown to affect Hmg1 regulation. Based on their overall findings, the authors suggest a second mechanistic action for triazoles through induction of negative feedback regulation on Hmg1 – hence, the title of the paper.

The research topic is important. The paper is well written. Experimental results are clearly and logically presented, and most conclusions are supported by the results. The major shortcomings of the manuscript being the conclusions are mostly confirmative (e.g. supporting what had been speculated previously) and many findings are similar to what had been found in yeasts or mammalian cells. More importantly, there is a lack of mechanistic insight into the negative feedback regulation of Hmg1. The authors proposed a few models based on findings in yeasts and mammalian cells and provided experimental evidence demonstrating that the models are not conserved in *Aspergillus fumigatus*. However, the molecular details of the negative feedback regulation remain largely unknown.

Major comments:

1) The authors suggested that the Hmg1 regulation and triazoles resistance is affected by the stoichiometric balance between Hmg1 and InsA, based on the findings in other organisms. However, it is not clear whether Hmg1 and InsA protein levels are different in the SSD mutants versus wildtype non-resistant strains and whether the levels change in the presence and absence of triazoles. These results would be needed for the claim.

2) Related to the proteins level, does Hmg1 protein level go down with InsA over-expression in wildtype and SSD mutants (and vice versa)? The finding will inform whether InsA action is mediated through controlling Hmg1 levels (e.g. degradation).

Minor comments:

3) Line 141-142: The authors stated that “we first sought to expand on our previous analysis by employing a genomic dataset from 87 globally distributed *A. fumigatus* clinical isolates”. The word “globally” is an overstatement, as the isolates are mainly coming from US and UK with four from two European countries and 2 from Japan.

4) Lines 173: “Depending on the triazole utilized, we noted 2- to 8-fold increased MIC values for the D242H, W272L, C402R, V403D, and G483R variants (Figure 2B-E).” How do these MIC values compare with those of the respective clinical isolates? If the MIC values are the same as clinical isolates, especially those that has both *hmg1* and *cyp51A/B* (or other) mutations, then the result lend additional support to the Hmg1-mediated mechanism is the key mechanism for triazole resistance.

5) Lines 249-251: “When compared to wild type and manipulation control strains, no change in susceptibility was evident in the presence of rosuvastatin for any Hmg1 SSD mutant strain (Figure 4B).” The *hmg1F262del* seems to be more sensitive than the control strains on rosuvastatin.

6) Fig. 2A: The S305P variant has been used in multiple experiments. It would be helpful to readers to indicate the position of the S305P residue on the schematic protein structure.

7) Fig. 3B: It is not clear what is being displayed – e.g. the color key does not have a unit. Based on the text and color scale (negative 4 to positive 2), it appears that the colors represent fold change. However, it is not clear how the fold change is calculated – e.g. what are the values comparing to (C under NT)? Or are the values z-scores? Please add unit to the color scale and clarify with details. The figure may also be improved by marking the genes in the Mevalonate and Ergosterol pathways – e.g. in the same colors as in Fig. 3A.

8) Fig. 4D and Supplementary data file (Clinical isolates information): There seems to be a discrepancy in the resistant concentration for VORI. The table lists in mg/ml, while the figures has µg/ml. Moreover, the excel shows that the Hmg1-S305P (DI-20-109) and Hmg1-C402R (DI-20-110) mutants have a VORI concentration as 4 mg/ml, but these mutants can barely grow on 0.5-1 µg/ml of Voriconazole. Is this a typo mistake in the concentrations? Please clarify.

9) Line 210-212 stated that "We also previously noted that the relative decrease in the proportion of ergosterol in mutant strains was not correlated with a decrease in the total ergosterol content (µg ergosterol/mg dry weight)". In Figure 3C, only relative proportions were shown. The absolute amount of sterol contents should also be presented to show the overall picture of the mutants' effects and to support the statements in lines 210 and 232 that the changes of total contents are not correlated with the relative proportion changes in the mutants.

10) Fig. 5E: Is there a mixed up for the photo of the hmg1pHspA-1 strain on 0.25 and 0.5 µg/ml Terbinafine? The growth on 0.5 appears to be better than that on 0.25 (i.e. colony size is bigger).

11) Fig 8B: It would be useful to include the hmg1 SSD mutants in this growth test to compare their resistance with that conferred by the loss of HrdA in the Δ hrdA mutant. The result would allow readers to judge whether SSD and Δ hrdA mutations have similar effects and actions.

12) Line 329: The experimental result for lanosterol accumulation should be presented, instead of referring to another work under submission without any reference.

13) Line 509, the "*A. fumigatus*" should be italicized.

14) Lines 604-606: "Whole-genome sequencing data files for the *A. fumigatus* isolates sequenced as part of this study have been deposited in NCBI SRA under the accession number XXXXXXXX. Transcriptome data have been deposited in GEO/NCBI under accession number XXXXX." Accession numbers are missing.

Reviewer #2:

Comment 1: In the Abstract and the main text (e.g., line 71) describe the action of azoles exclusively as "ergosterol depleting" and consequently their identified mechanism several times in the manuscript as "novel secondary mechanism" (e.g., line 43, 50, 135, etc). However, it has been shown in several fungal species that azole treatment results not only in depletion of ergosterol but concomitantly to accumulation of toxic pathway intermediates that disrupt the cell membrane (PMID: 25320648). This has been neglected in this manuscript and should be complemented.

Response: We thank the reviewer for this comment and apologize for overlooking this issue. We have now included this information in the Introduction and added the suggested reference (Lines 71-72). We also included this content in the revised Discussion (Lines 494-497).

Comment 2: Recent studies revealed that there are two to three primary populations/clades in *A. fumigatus* (PMID: 36395320, PMID: 35469019). It would be interesting to know if the azole resistance causing mutations within HMGCR are population/clade specific or not.

Response: We have now included an updated Figure 1 that displays the clades into which our clinical isolates fall. These analyses are based on the studies noted by the reviewer published in Nature Microbiology (PMID: 35469019). We note that many of the isolates in our collection that encode Hmg1 SSD mutations fall within Clade A. However, this is not an exclusive relationship, as multiple isolates within Clade B also carry SSD mutations. This information is now noted at Lines 155-168.

Comment 3: Line 185: Which mutant strains? I assume CEA10 strains with introduced mutations.

Response: Yes. We have updated the text for greater clarity (Lines 203-204).

Comment 4: Line 192: "upregulated in common" compared to what?; this page: wouldn't it be interesting to comment also the downregulated genes?

Response: We have clarified in the text that these comparisons were made to the control strain at Lines 210, 212, and 219. We have also now included a section highlighting the genes downregulated in common (Lines 225-239) and have complemented this with the addition of data to Supplemental File 2.

Comment 5: Line 276: The fact that overexpression of *hmg1* increases resistance to triazoles but not terbinafine has been reported previously (PMID: 36350143).

Response: We have updated to text to reference the indicated manuscript (Line 314).

Comment 6: Figure 7D: the strain *hmg1*(WT)*erg6*(pTetOff) shows decreased growth without doxycycline (most upper row) compared to *hmg1*(WT), which might be caused by overexpression of *erg6* as this is driven by unrepressed pTetOff. If this holds true, this should result in lanosterol depletion. It might be interesting to assay triazole susceptibility of strain *hmg1*(WT)*erg6*(pTetOff) compared to *hmg1*(WT).

Response: We see the reviewer's point. To make this comparison using the original figure is not completely valid, as we did not quantitate colony diameters. We have now included new data

showing, quantitatively, that the hmg1(WT)erg6(pTetOff) mutant does not display decreased growth without doxycycline compared to the hgm1(WT) strain and that this genetic manipulation does not alter triazole susceptibility. This data is included in the newly added Supplementary Figure 7 and is referenced in Lines 368-371.

Comment 7: In experiments using the pTetOff promoter, the effect of doxycycline on expression of the respective gene should be shown, at least exemplary.

Response: We have now included this data in the new Supplementary Figure 7 and have updated the text to reference this new data (Lines 368-371).

Reviewer #3:

Comment 1: The Introduction wanders at times and should be condensed. At times, it appears more like a Discussion than an Introduction. These wanderings detract from the impending objective, especially for non-specialists.

Response: We appreciate the Reviewer's desire to focus the Introduction on the most salient points. Therefore, we have removed any elements that do not directly relate to the problem of triazole resistance and the role that HMGCR mutations may play and were intended for study in this manuscript.

Comment 2: Lines 142. Can you clarify the origin of the global clinical isolates obtained. It is stated in the Methods that isolates were obtained from the Fungus Testing Lab and represented newly acquired US isolates. This might be somewhat problematic as US isolates are rarely drug resistant. Please clarify origin.

Response: As stated in the Methods section, the only physically newly acquired isolates used for this study were from the Fungus Testing Laboratory or the CDC in the US. The additional publicly available genome and susceptibility data used herein to broaden our comparisons were from isolates originally reported in England, Ireland, Wales, Netherlands, Germany and Japan. All of this information is listed in Supplementary File 1 and is referenced in the text (Lines 143-145).

Comment 3: Line 173. The MIC ranges are not very much higher than susceptible strains (MIC 2-8 fold). How do the authors know that these isolates are drug resistant and not drug tolerant? The former would be untreated or poorly treated in animal models and would be expected to induce a higher probability for clinically refractive responses. The authors might want to consider that while this new mechanism may elevate MIC, it might induce resistance in some cases but may be more important for tolerance development. Such tolerance would predispose such isolates downstream for generating Cyp51A mutations. Have the authors observed SSD mutations in isolates with Cyp51A mutations?

Response: We thank the reviewer for this comment. While this is speculative, it is certainly an issue that the authors have discussed and considered. Figure 1 clearly shows that many of the isolates we identified as carrying Hmg1 SSD mutations also carry non-synonymous SNPs in the *cyp51A* coding region. Since our work suggests that negative feedback on HMGCR is a secondary MOA for the normal anti-*Aspergillus* activity of triazoles, it would make sense that Hmg1 SSD mutations could arise to promote low-level drug resistance (or even tolerance) and subsequently potentiate target gene mutation for selection of high-level resistance. Although this is not expected to be an essential first step, it is certainly an interesting and logical speculation. We have updated the Discussion to include this point (Lines 535-539).

Reviewer #4:

Comment 1: The authors suggested that the Hmg1 regulation and triazoles resistance is affected by the stoichiometric balance between Hmg1 and InsA, based on the findings in other organisms. However, it is not clear whether Hmg1 and InsA protein levels are different in the SSD mutants versus wildtype non-resistant strains and whether the levels change in the presence and absence of triazoles. These results would be needed for the claim.

Response: We thank the reviewer for this comment focusing on measuring protein abundance as an important aspect of delineating the SSD-mediated triazole resistance phenotype. To clarify, we do not assert that Hmg1 SSD mutations would impact InsA protein levels. To our knowledge, there is no model proposed in the literature in which HMGCR imparts control over INSIG via regulation of INSIG protein levels. The protein degradation-based mechanism that is described in the scientific literature only points to control of HMGCR abundance in response to sterol accumulation in mammals. For this mechanism, INSIG binds HMGCR in the presence of sterols and promotes HMGCR degradation via an E3-ubiquitin ligase that ubiquitinates HMGCR for degradation. We feel this is clearly stated in our manuscript in text and graphically in Figure 8. Our assertion is simply that the mutational analyses we completed, which are expected to result in alteration of the protein levels of either Hmg1 or InsA (i.e., our overexpression mutants), generated results that are supportive of a conserved INSIG-HMGCR regulatory relationship in *A. fumigatus* as is seen in mammalian systems. The choice of the term “stoichiometric balance” was incorrect on our part. Its use implies that there exists normal back-and-forth regulation between INSIG-HMGCR proteins to maintain homeostasis. As described above and in our manuscript, regulation of protein abundance is only imparted onto the HMGCR protein through this mechanism. To clarify, we have removed the term “stoichiometric balance” in multiple sections of the manuscript.

The underlying question, however, is an important one. Do SSD mutations affect hmg1 protein abundance in response to triazole stress? To answer this question, we first attempted multiple rounds of western blot analyses using multiple epitope tags, as well as a custom developed anti-*hmg1* antibody, to measure Hmg1 protein levels in response to drug stress. These experiments were unsuccessful due to low level expression of the *hmg1* gene in *A. fumigatus*. As an alternative approach, we generated new Hmg1 GFP-chimeric strains in both the Hmg1-WT and Hmg1-S305P genetic backgrounds with the aim of employing fluorescence readouts of lysates to quantitate concentration of GFP as a proxy for Hmg1 protein levels. Expression of the Hmg1-GFP proteins revealed an expected localization to the peri-nuclear ER and confirmed that Hmg1 protein is expressed at low levels under normal conditions. After validating that the C-terminal GFP fusion did not affect growth, drug susceptibility, or the impact of the SSD mutation (new Figure 9) and that our assay reliably measured GFP protein levels in *A. fumigatus* lysates (new Supplemental Figure 8), we assayed both the control and Hmg1 SSD mutant for alteration in Hmg1 protein levels in response to voriconazole stress. The new results reveal that triazole stress does not alter localization or decrease the abundance of Hmg1 (new Figure 10). The presentation of data is found at Lines 428-472. These new results correlate well with our genetic evidence provided in Figure 8.

Comment 2: Related to the proteins level, does Hmg1 protein level go down with InsA over-expression in wildtype and SSD mutants (and vice versa)? The finding will inform whether InsA action is mediated through controlling Hmg1 levels (e.g. degradation).

Response: We thank the reviewer again for the focus on quantifying protein abundance changes to support our genetic evidence. As explained above, our new analyses now confirm

that the triazole-induced, INSIG-mediated negative feedback of Hmg1 is not orchestrated through down-regulation of Hmg1 protein abundance, Therefore, we have not completed the additional mutational analyses for *insA* overexpression in our *hmg1-gfp* mutant backgrounds.

Comment 3: Line 141-142: The authors stated that “we first sought to expand on our previous analysis by employing a genomic dataset from 87 globally distributed *A. fumigatus* clinical isolates”. The word “globally” is an overstatement, as the isolates are mainly coming from US and UK with four from two European countries and 2 from Japan.

Response: We appreciate the reviewer’s comment and have modified this statement for clarity (Lines 143-145).

Comment 4: Lines 173: “Depending on the triazole utilized, we noted 2- to 8-fold increased MIC values for the D242H, W272L, C402R, V403D, and G483R variants (Figure 2B-E).” How do these MIC values compare with those of the respective clinical isolates? If the MIC values are the same as clinical isolates, especially those that has both *hmg1* and *cyp51A/B* (or other) mutations, then the result lend additional support to the Hmg1-mediated mechanism is the key mechanism for triazole resistance.

Response: We have updated the text to discuss this point. As can be seen in Figure 1, of the isolates that carry the indicated *hmg1* SSD mutations, DI-20-118, DI-20-80, DI-20-135, C117, DI-20-84, and DI-20-129 each also have non-synonymous SNPs within the *cyp51A* coding region. Although we cannot rule out a role for these target gene mutations in driving resistance (or, other previously uncharacterized mechanisms for that matter), we now point out that six of the eight represented isolates carrying the noted *hmg1* mutations have triazole MIC values that are largely within a single dilution of those we report for our mutant strains. Therefore, the *hmg1* mutations within those isolates likely account for the majority of the resistance detected. Line 195-198.

Comment 5: Lines 249-251: “When compared to wild type and manipulation control strains, no change in susceptibility was evident in the presence of rosuvastatin for any Hmg1 SSD mutant strain (Figure 4B).” The *hmg1F262del* seems to be more sensitive than the control strains on rosuvastatin.

Response: To denote hypersusceptibility, we are using 100% growth inhibition as an endpoint. This is standard for fungicidal agents against *Aspergillus*. However, we understand that this is a stringent endpoint and that smaller alterations in susceptibility can be discerned by quantifying changes in colony diameter (or, growth rate) in the presence of drug. It is in entirely possible that each of the individual SSD mutations could have slightly different effects for a variety of reasons. We have modified the text at line 287 to clarify this point.

Comment 6: Fig. 2A: The S305P variant has been used in multiple experiments. It would be helpful to readers to indicate the position of the S305P residue on the schematic protein structure.

Response: Figure 2A has been updated to include the S305P mutation.

Comment 7: Fig. 3B: It is not clear what is being displayed – e.g. the color key does not have a unit. Based on the text and color scale (negative 4 to positive 2), it appears that the colors represent fold change. However, it is not clear how the fold change is calculated – e.g. what are the values comparing to (C under NT)? Or are the values z-scores? Please add unit to the

color scale and clarify with details. The figure may also be improved by marking the genes in the Mevalonate and Ergosterol pathways – e.g. in the same colors as in Fig. 3A.

Response: We apologize for this oversight. The reviewer is correct, the Figure depicts fold change (log₂) and all values are compared to the NT or voriconazole treated control. We have updated the figure and figure legend for greater clarity.

Comment 8: Fig. 4D and Supplementary data file (Clinical isolates information): There seems to be a discrepancy in the resistant concentration for VORI. The table lists in mg/ml, while the figures has µg/ml. Moreover, the excel shows that the Hmg1-S305P (DI-20-109) and Hmg1-C402R (DI-20-110) mutants have a VORI concentration as 4 mg/ml, but these mutants can barely grow on 0.5-1 µg/ml of Voriconazole. Is this a typo mistake in the concentrations? Please clarify.

Response: We apologize for this oversight. In the Supplementary File, “mg/ml” should have read “mg/L”. This was a typo. We have corrected the File to reflect this. As to the second point, this is actually not a discrepancy at all. The Supplementary File displays values for clinical isolates in which each of these hmg1 mutations were noted. Figure 4D (and all of the Figures here) show results for mutant strains we constructed in the laboratory so that we could isolate the effects of the individual hmg1 mutations for study.

Comment 9: Line 210-212 stated that “We also previously noted that the relative decrease in the proportion of ergosterol in mutant strains was not correlated with a decrease in the total ergosterol content (µg ergosterol/mg dry weight)”. In Figure 3C, only relative proportions were shown. The absolute amount of sterol contents should also be presented to show the overall picture of the mutants’ effects and to support the statements in lines 210 and 232 that the changes of total contents are not correlated with the relative proportion changes in the mutants.

Response: To support this statement, we have now included the analysis to calculate total ergosterol content as we previously described. As can be seen from our analysis, the relative decrease in the proportion of ergosterol in the mutant strains was not associated with a decrease in the µg ergosterol / mg of dry weight. We have updated the Sterol Profile excel file and have updated the text to reflect this (Lines 267-269).

Comment 10: Fig. 5E: Is there a mixed up for the photo of the hmg1^{pHspA-1} strain on 0.25 and 0.5 µg/ml Terbinafine? The growth on 0.5 appears to be better than that on 0.25 (i.e., colony size is bigger).

Response: In brief, no, the images are not mixed up. The uncropped image of these plates was provided in the Supplementary Figure 4. In comparing differences in susceptibility for the data presented here, we are largely focusing on the drug concentration at which there is zero fungal growth. This 100% inhibition of growth would be the “minimum inhibitory concentration” of a fungicidal compound against *Aspergillus*. That said, we do take the reviewers point and agree that the two colonies for this one strain at 0.5 and 0.25 µg/ml terbinafine are not much different in growth. The colony at 0.5 µg/ml does appear slightly larger, however, we have not quantified colony diameter here. There is always some variability in resulting colony diameters when these assays are run (especially when drug stress is applied), due to small variations in inoculum or drug implanted in the media. This is one of the reasons we provided data from multiple strains for both the overexpression mutants used in this figure. As can be seen for the *hmg1*^{pHspA-2} mutant strain (directly below the strain indicated by the reviewer, on the same figure) the colony sizes are as expected in respect to drug. This is simple biological and technical variation.

Comment 11: Fig 8B: It would be useful to include the *hmg1* SSD mutants in this growth test to compare their resistance with that conferred by the loss of HrdA in the Δ *hrdA* mutant. The result would allow readers to judge whether SSD and Δ *hrdA* mutations have similar effects and actions.

Response: Using the same experimental approach (e.g., media composition, fungal inoculum, time and temperature of incubation, etc.), the effect of voriconazole on growth of the *hmg1* SSD mutants can be seen in Figure 4 and Figure 6. Although these experiments were completed on different days in different “batches”, the *hmg1*^{S305P} mutant shows exactly the same result of near complete loss of growth at 1 μ g / ml voriconazole (and, no growth at 2 μ g / ml). Therefore, the results achieved with voriconazole are consistently replicated. We feel it is clear from comparing Figures 4 and 6 to Figure 8 that the *hrdA* deletion mutant is within a single two-fold dilution of the *hmg1* SSD mutants with respect to voriconazole susceptibility.

Comment 12: Line 329: The experimental result for lanosterol accumulation should be presented, instead of referring to another work under submission without any reference.

Response: While the data related to this study are currently under revision after review for separate publication in Nature Communications, we have submitted this study as a pre-print to BioRxiv and we have now updated the text with this reference (Line 375).

Comment 13: Line 509, the “*A. fumigatus*” should be italicized.

Response: Corrected.

Comment 14: Lines 604-606: “Whole-genome sequencing data files for the *A. fumigatus* isolates sequenced as part of this study have been deposited in NCBI SRA under the accession number XXXXXXXX. Transcriptome data have been deposited in GEO/NCBI under accession number XXXXX.” Accession numbers are missing.

Response: The whole genome and transcriptome sequencing data file have been uploaded to SRA and the accession numbers updated in the text. We have also updated the whole genome sequence biosample information for each isolate in the supplementary files.

REVIEWERS' COMMENTS

Reviewer #2 (Remarks to the Author):

All my questions and concerns have been adequately addressed in the revised version. I congratulate the authors on their study!

Reviewer #3 (Remarks to the Author):

The authors have been highly responsive to all of the reviewers comments, which is most appreciated. Text clarifications, new data, and expanded discussion and interpretations have been well handled in the revised manuscript, which is now suitable for publication.

Reviewer #4 (Remarks to the Author):

The authors have addressed my comments.